# MemBench: Memorized Image Trigger Prompt Dataset for Diffusion Models

**Chunsan Hong**         *hoarer@kaist.ac.kr*
*KAIST*
*School of Electrical Engineering*

**Tae-Hyun Oh**[†]         *thoh.kaist.ac.kr@gmail.com*
*KAIST*
*School of Computing*

**Minhyuk Sung**[†]         *mhsung@kaist.ac.kr*
*KAIST*
*School of Computing*

**Reviewed on OpenReview:** *https://openreview.net/forum?id=E78OaH2s3f&noteId=GpBYx75flr*

## Abstract

Diffusion models have achieved remarkable success in Text-to-Image generation tasks, leading to the development of many commercial models. However, recent studies have reported that diffusion models often repeatedly generate memorized images in train data when triggered by specific prompts, potentially raising social issues ranging from copyright to privacy concerns. To sidestep the memorization, recent studies have been conducted to develop memorization mitigation methods for diffusion models. Nevertheless, the lack of benchmarks hinders the assessment of the true effectiveness of these methods. In this work, we present MemBench, the first benchmark for evaluating image memorization mitigation methods. Our benchmark includes a large number of memorized image trigger prompts in various Text-to-Image diffusion models. Furthermore, in contrast to the prior work evaluating mitigation performance only on trigger prompts, we present metrics evaluating on both trigger prompts and general prompts, so that we can see whether mitigation methods address the memorization issue while maintaining performance for general prompts. Through our MemBench evaluation, we revealed that existing memorization mitigation methods notably degrade overall performance of diffusion models and need to be further developed. The code and datasets are available at https://github.com/chunsanHong/MemBench_code

## 1 Introduction

Text-to-Image (T2I) generation has shown significant advancements and successes with the advance of diffusion models. Compared to previous generative models, text-conditional diffusion models excel in generating diverse and high quality images from user-desired text prompts, which has led to the vast release of commercial models such as MidJourney. However, recent studies (Somepalli et al., 2023a;b; Carlini et al., 2023) have revealed that certain text prompts tend to keep replicating images in the train dataset which can cause private data leakage leading to potentially serious privacy issues. This issue has already triggered controversy in the real world: specific prompts containing the term "Afghan" have been known to reproduce copyrighted images of the Afghan girl when using MidJourney (Wen et al., 2024b). One of the major issues with such prompts is that, regardless of initial random noise leveraged in the reverse process of the diffusion model, they always invoke almost or exactly same memorized images (Wen et al., 2024b; Carlini et al., 2023; Webster, 2023).

---

[†]Co-corresponding authors.

To address this matter, Wen et al. (2024b) and Somepalli et al. (2023b) have proposed mitigation methods to prevent the regeneration of identical images in the train dataset invoked from certain text prompts. However, the evaluation of these memorization mitigation methods has lacked rigor and comprehensiveness due to the absence of benchmarks. As an adhoc assessment method, the current studies (Wen et al., 2024b; Somepalli et al., 2023b) have adopted the following workaround: 1) simulating memorization by fine-tuning T2I diffusion models for overfitting on a separate small and specific dataset of {image, prompt} pairs, and 2) assessing whether the images used in the fine-tuning are reproduced from the query prompts after applying mitigation methods. However, it remains unclear whether such results can be extended to practical scenarios with the existing large-scale pre-trained diffusion models and can represent the effectiveness for resolving memorization.

In this work, we present **MemBench**, the first benchmark for evaluating image memorization mitigation methods for diffusion models. Our MemBench includes the following key features to ensure effective evaluation: (1) MemBench provides 3,000, 1,500, 309, and 1,352 memorized image trigger prompts for Stable Diffusion 1, 2, DeepFloydIF (Shonenkov et al., 2023), and Realistic Vision (CivitAI, 2023), respectively. In contrast, previous work (Webster, 2023) only provided 325, 210, 162, and 354 prompts. By increasing the number of prompts, we enhance the reliability of the evaluation. (2) We take into account a general prompt scenario to assess the side-effects of mitigation methods, which has been overlooked in prior work. The prior mitigation methods (Wen et al., 2024b; Ren et al., 2024; Somepalli et al., 2023b) have been evaluated solely on memorized image trigger prompts, but has often exposed the side-effect of performance degrading. Ideally, the performance on general prompts should be maintained even after mitigation methods are deployed. (3) We suggest to use multiple metrics. As previous mitigation works (Wen et al., 2024b; Somepalli et al., 2023b) have measured, MemBench includes SSCD (Pizzi et al., 2022), which measures the similarity between memorized and generated images, and CLIP Score (Hessel et al., 2021), which measures Text-Image alignment. Additionally, MemBench involves Aesthetic Score (Schuhmann et al., 2022) to assess image quality, which has been overlooked by prior work and allows to penalize unuseful trivial solutions. (4) We propose the reference performance that mitigation methods should achieve to be considered effective. In previous works (Ren et al., 2024; Wen et al., 2024b; Somepalli et al., 2023b), the effectiveness of mitigation methods has been demonstrated by measuring the decrease in SSCD and the extent to which the CLIP Score is maintained before and after applying the mitigation method. However, this does not necessarily confirm whether image memorization has been adequately mitigated. Therefore, we provide guidelines on the target values.

**Benchmark Contributions.** We observe several limitations of current mitigation methods by evaluating those in MemBench, which consists of large test data, diverse metrics, and scenarios. We confirm that mitigation methods decrease Text-Image alignment when applied to the image generation of trigger prompts. We found that even when accepting a significant sacrifice in Text-Image alignment and applying a mitigation method with very high strength, its SSCD value still fails to reach the reference performance we proposed. Furthermore, we reveal a new observation that these methods significant decrease Aesthetic Score, which highlights the generation of low-quality images. We clarify that while previous work (Ren et al., 2024) measured FID, FID does not effectively measure image quality. We also found that in addition to lower general quality, the variance in the quality of generated images increases. In the general prompt scenario, mitigation methods degrade generation performance, making practical application difficult.

**Technical Contributions.** Our additional contribution lies in offering an effective algorithm to search for memorized image trigger prompts. The absence of such benchmarks originates from the significant challenge of collecting prompts that induce memorized images. Existing searching methods (Carlini et al., 2023; Webster et al., 2023) require extensive computational resources, large system memory, and access to the diffusion model's training data to function. Furthermore, with the LAION dataset now private[1], these methods have become unusable. In contrast, our proposed searching algorithm, based on Markov Chain Monte Carlo (MCMC), offers a more efficient approach to searching for problematic prompts directly within an open token space, without relying on any dataset. Notably, our method is currently the only available approach that can operate under these constraints.

**Empirical Findings.** We analyzed the memorized images found by our method and confirmed that these include commercial products that are actually being sold. Furthermore, we provide a possible explanation for

---

[1]https://laion.ai/notes/laion-maintenance/

why diffusion models trained on LAION-5B, where image deduplication has been applied, still exhibit image memorization: layout duplication.

## 2 Related Work

**Memorization in Diffusion Models.** Recently, diffusion models have been reported to replicate the images in the train dataset (Somepalli et al., 2023a;b; Carlini et al., 2023). Empirically, previous works (Somepalli et al., 2023b; Gu et al., 2023) attribute image memorization to duplicated images in the training data. Biroli et al. (2024) theoretically analyze the training dynamics of diffusion models and suggest that memorization arises from transitions into a feature-space fixed point regime.

**Memorization Mitigation Methods.** Memorization mitigation methods are divided into two categories: the inference time methods and the training time methods. The inference time methods aim to prevent the generation of images that are already memorized in pretrained diffusion models during the generation process. Somepalli et al. (2023b) propose a rule-based text embedding augmentation to mitigate memorization. This includes adding Gaussian noise to text embeddings or inserting random tokens in the prompt. Wen et al. (2024b) propose a loss that predicts if a prompt will induce a memorized image, and present a mitigation strategy that applies adversarial attacks on this loss to modify the text embeddings of trigger prompts. Both of these works evaluate their methods by intentionally overfitting the diffusion model on specific small {image, text} pairs to induce the memorization effect, and then checking whether the images are regenerated from the corresponding prompts when their methods are applied. Ren et al. (2024) analyze the impact of trigger prompts on the cross-attention layer of diffusion models and propose a corresponding mitigation method.

Train time methods aim to prevent diffusion models from memorizing training data during model training by employing specific training techniques. Although several methods (Daras et al., 2024; Liu et al., 2024) have been proposed, experiments have been conducted only on small models and datasets such as CIFAR-10 and CelebHQ. While some experiments (Ren et al., 2024; Wen et al., 2024b) have been conducted on large models such as Stable Diffusion, they only assess whether the fine-tuning dataset is memorized when fine-tuning the model. To date, no train time mitigation method has been tested by training large-scale diffusion models from scratch to evaluate its effectiveness.

In this work, we focus on the inference time methods, considering the practical scenarios of utilizing existing large-scale pre-trained diffusion models, such as Stable Diffusion. To effectively evaluate these methods, we introduce MemBench, which provides sufficient test data and appropriate metrics for comprehensive assessment.

**Training Data Extraction Attack.** Our MemBench is constructed by our proposed computational method that shares a similar vein with the following attack methods. Carlini et al. (2023) propose a method to search for memorized image trigger prompts in Stable Diffusion. In the pre-processing stage, they embed the entire training set of Stable Diffusion into the CLIP (Radford et al., 2021) feature space and cluster these embeddings to identify the most repeated images. In the post-processing stage, Stable Diffusion is used to generate 500 images for each prompt corresponding to these clustered images. The similarity among these 500 generated images is measured, and only those prompts that produce highly similar images are sampled. Finally, image retrieval is performed on the training data using generated images from these selected prompts to verify if the generated images match the training data images. The pre-processing involves CLIP embedding and clustering of 160M images, while the post-processing involves generating 175M images, *i.e.*, computationally demanding. Webster (2023) propose an advanced searching algorithm. In the pre-processing stage, an encoder is trained to compress CLIP embeddings. Then, 2B CLIP embeddings are compressed and clustered using KNN (Webster et al., 2023). In the post-processing stage, Webster introduces an effective method that performs a few inferences of the diffusion model to predict whether a prompt will induce memorized images. This method is applied to 20M prompts acquired from the pre-processing stage.

Both methods share common bottlenecks: they are memory inefficient and require extremely high computational costs. Moreover, the most fundamental problem is their reliance on training data as candidate trigger prompts. With LAION becoming inaccessible[1], these methods can no longer be reproducible and utilized.

However, our method can search more for trigger prompts efficiently than those methods even without any pre-processing steps and any dataset.

In another line of research, Chen et al. (2024) propose a method for extracting training data from unconditional diffusion models. In contrast, several studies (Somepalli et al., 2023b; Gu et al., 2023) indicate that conditioning plays a critical role in memorization, with unconditional models being less susceptible to it. Furthermore, since T2I diffusion models are the ones widely applied in real-world scenarios, our work focuses on constructing a memorization benchmark for T2I diffusion models.

**Benchmark Dataset.** Since the only existing dataset that can be used for evaluating mitigation methods is the small dataset released by Webster (2023), Ren et al. (2024) evaluate their method on the Webster dataset, while it is not originally purposed as a benchmark dataset. The dataset is constructed by the training data extraction attack method proposed by Webster, which is not scalable; thus, the dataset remains a small scale. Also, Ren et al. did not measure the loss of semantic preservation after mitigation, which is an important criterion but overlooked. Our benchmark is the first benchmark for evaluating those mitigation methods with carefully designed metrics and sufficient test data.

## 3 Searching Memorized Image Trigger Prompt with MCMC

We present our proposed scalable computational method to construct our MemBench dataset. Given a pre-trained diffusion model, we computationally search memorized image trigger text prompt. In this section, we first brief the preliminaries, formulate the search as an optimization problem, and propose a Markov Chain Monte Carlo algorithm.

### 3.1 Preliminary

**Diffusion Models.** Denoising Diffusion Probabilistic Model (DDPM) (Ho et al., 2020) is a representative diffusion model designed to approximate the real data distribution $q(\mathbf{x})$ with a model $p_{\boldsymbol{\theta}}(\mathbf{x})$. For each $\mathbf{x}_0 \sim q(\mathbf{x})$, DDPM constructs a discrete Markov chain $\{\mathbf{x}_0, \mathbf{x}_1, \ldots, \mathbf{x}_T\}$ that satisfies $q(\mathbf{x}_t|\mathbf{x}_{t-1}) = \mathcal{N}(\mathbf{x}_t; \sqrt{1-\beta_t}\mathbf{x}_{t-1}, \beta_t\mathbf{I})$. This is referred to as the forward process, where $\{\beta_t\}_{t=1}^T$ is a sequence of positive noise scales. Conversely, the reverse process generates images according to $p_{\boldsymbol{\theta}}(\mathbf{x}_{t-1}|\mathbf{x}_t) = \mathcal{N}(\mathbf{x}_t; \mu_{\boldsymbol{\theta}}(\mathbf{x}_t, t), \Sigma_{\boldsymbol{\theta}}(\mathbf{x}_t, t))$. DDPM starts by sampling $\mathbf{x}_T$ from a Gaussian distribution, and then undergoes a stochastic reverse process to generate the sample $\mathbf{x}_0$, *i.e.* an image. With a parametrized denoising network $\boldsymbol{\epsilon}_{\boldsymbol{\theta}}$, this generation process can be expressed as:

$$\mathbf{x}_{t-1} = \frac{1}{\sqrt{\alpha_t}}\left(\mathbf{x}_t - \frac{1-\alpha_t}{\sqrt{1-\bar{\alpha}_t}}\boldsymbol{\epsilon}_{\boldsymbol{\theta}}(\mathbf{x}_t, t)\right) + \sigma_t\mathbf{w}, \tag{1}$$

where $\alpha_t = 1 - \beta_t$, $\bar{\alpha}_t = \prod_{i=1}^t \alpha_t$, $\sigma_t$ can be $\sqrt{\beta}$ or $\sqrt{\frac{1-\bar{\alpha}_{t-1}}{1-\bar{\alpha}_t}\beta_t}$, and $\mathbf{w} \sim \mathcal{N}(0; \mathbf{I})$. The equations may vary depending on hyper-parameter choices and the numerical solver used (Song et al., 2021a;b).

**Classifier Free Guidance (CFG).** In T2I diffusion models such as Stable Diffusion (Rombach et al., 2022), CFG (Ho & Salimans, 2022) is commonly employed to generate images better aligned with the desired prompt. Given a text prompt $\mathbf{p}$ and the text encoder $\mathbf{f}(\cdot)$ of the pre-trained CLIP (Radford et al., 2021), predicted noise is replaced as follows:

$$\tilde{\boldsymbol{\epsilon}}_{\boldsymbol{\theta}, \mathbf{f}}(\mathbf{x}, \mathbf{p}, t) = \boldsymbol{\epsilon}_{\boldsymbol{\theta}}(\mathbf{x}, \mathbf{f}(\emptyset), t) + s \cdot (\boldsymbol{\epsilon}_{\boldsymbol{\theta}}(\mathbf{x}, \mathbf{f}(\mathbf{p}), t) - \boldsymbol{\epsilon}_{\boldsymbol{\theta}}(\mathbf{x}, \mathbf{f}(\emptyset), t)), \tag{2}$$

where $\emptyset$ denotes the empty string, and $s$ is the guidance scale.

**Image Memorization and Similarity Measurement Score.** We follow the definition of image memorization suggested by Carlini et al. (2023):

**Definition 1 ($\tau$-Image Memorization)** *Given a train set $\mathcal{D}_{train} = \{(\mathbf{x}_{train,i}, \mathbf{p}_{train,i})\}_{i=1}^N$, a generated image $\mathbf{x}$ from a diffusion model $\boldsymbol{\epsilon}_{\boldsymbol{\theta}}$ trained on $\mathcal{D}_{train}$, and a similarity measurement score $\rho$, image memorization of $\mathbf{x}$ is defined as:*

$$\mathcal{M}_\tau(\mathbf{x}, \mathcal{D}_{train}) = \mathbb{I}\left[\exists \mathbf{x}_{train} \in \mathcal{D}_{train} \ s.t. \ \rho(\mathbf{x}, \mathbf{x}_{train}) > \tau\right], \tag{3}$$

*where $\tau$ is a threshold, $\mathbb{I}$ is indicator function, and $\mathcal{M}(\cdot)$ indicates whether the image is memorized.*

As in prior works (Wen et al., 2024b; Ren et al., 2024; Somepalli et al., 2023b), we utilize SSCD (Pizzi et al., 2022) score as a similarity measurement score, $\rho$. SSCD is a model designed to identify copied or manipulated images by learning robust image representations through self-supervised learning. SSCD score is defined as the cosine similarity between the features extracted from two images using the SSCD model. The model ensures effective image copy detection across diverse scenarios, such as cropping or filtering.

**Memorized Image Trigger Prompt Prediction.** Wen et al. (2024b) proposed an efficient method to predict whether a prompt will generate an image included in the training data. The prior works found that prompts inducing memorized images do so regardless of the initial noise, $\mathbf{x}_T$ (Carlini et al., 2023; Webster, 2023), *i.e.*, repeatedly generating the same or almost identical images despite different $\mathbf{x}_T$. To quickly identify this case, Wen et al. (2024b) propose a measure to predict whether a prompt will induce a memorized image using only the first step of the diffusion model, without generating the image. This measure, referred to as $d_{\boldsymbol{\theta}}$, is formulated as follows:

$$d_{\boldsymbol{\theta}}(\mathbf{p}) = \mathbb{E}_{\mathbf{x}_T \sim \mathcal{N}(0,\mathbf{I})}[||\boldsymbol{\epsilon}_{\boldsymbol{\theta}}(\mathbf{x}_T, \mathbf{f}(\mathbf{p}), T) - \boldsymbol{\epsilon}_{\boldsymbol{\theta}}(\mathbf{x}_T, \mathbf{f}(\emptyset), T)||_2]. \tag{4}$$

In this context, the larger $D_{\boldsymbol{\theta}}(\mathbf{p})$, the higher the probability that the image generated by the prompt is included in the training data. Denoting image $\mathbf{x}$ generated from diffusion model $\boldsymbol{\epsilon}_{\boldsymbol{\theta}}$ with prompt $\mathbf{p}$ as $\mathbf{x}(\boldsymbol{\epsilon}_{\boldsymbol{\theta}}, \mathbf{p})$, we re-purpose it by expressing as $d_{\boldsymbol{\theta}}(\mathbf{p}) \propto \mathbb{E}[\mathcal{M}(\mathbf{x}(\boldsymbol{\epsilon}_{\boldsymbol{\theta}}, \mathbf{p}), \mathcal{D}_{train})]$, where we omit $\tau$ for simplicity. To validate the effectiveness of detecting whether a prompt is a memorized image trigger prompt, Wen et al. construct a dataset containing memorized prompts provided by Webster (2023) and non-memorized prompts from LAION (Schuhmann et al., 2022), COCO (Lin et al., 2014), lexica.art (Santana, 2022) and randomly generated strings. The reported area under the curve (AUC) of the receiver operating characteristic (ROC) curve is 0.960 and 0.990 when the number of initial noises is 1 and 4, respectively.

## 3.2 Memorization Trigger Prompt Searching as an Optimization Problem

Our objective is to construct a memorized image trigger prompts dataset and verify corresponding memorized images, *i.e.* to construct $\mathcal{D}_{mem} = \{\mathbf{p} \mid \mathbb{E}[\mathcal{M}(\mathbf{x}(\boldsymbol{\epsilon}_{\boldsymbol{\theta}}, \mathbf{p}), \mathcal{D}_{train})] > \kappa, \mathbf{p} \in \mathcal{T}\}$ where $\kappa$ is the threshold and $\mathcal{T}$ is space of all possible prompts. As mentioned in Section 2, the prior works (Carlini et al., 2023; Webster, 2023) utilized $\mathcal{D}_{train}$ to search for candidate prompts that could become $\mathcal{D}_{mem}$. They then generated images from these candidate prompts and conducted image retrieval to find memorized images within $\mathcal{D}_{train}$ which is expensive. Moreover, since the training dataset, LAION, is no longer accessible, this approach becomes infeasible. Thus, we approach the problem from a different perspective. We search for candidate prompts that could become $\mathcal{D}_{mem}$ without using $\mathcal{D}_{train}$. Then, we generate images from these candidate prompts and use a Reverse Image Search API[2] to find images on the web akin to generated ones by regarding the web as the training set. Finally, we perform a human verification process.

Given that $d_{\boldsymbol{\theta}}(\mathbf{p}) \propto \mathbb{E}[\mathcal{M}(\mathbf{x}(\boldsymbol{\epsilon}_{\boldsymbol{\theta}}, \mathbf{p}), \mathcal{D}_{train})]$, constructing $\mathcal{D}_{mem}$ can be conceptualized as an optimization problem where we treat the prompt space as a reparametrization space and aim to find prompts yielding high $d_{\boldsymbol{\theta}}(\mathbf{p})$. To formulate the optimization problem, we define the prompt space. Given a finite set $\mathcal{W}$ containing all possible words (tokens), where $|\mathcal{W}| = m$, we model a sentence $\mathbf{p}$ with $n$ words as an ordered tuple drawn from the Cartesian product of $\mathcal{W}$, represented as $\mathcal{P} = \mathcal{W}^n$. To solve the optimization problem, we treat $d_{\boldsymbol{\theta}}(\cdot)$ as a negative energy function and model the target Boltzmann distribution $\pi$ such that higher values of $d_{\boldsymbol{\theta}}(\cdot)$ correspond to higher probabilities as

$$\pi(\mathbf{p}) = \frac{e^{d_{\boldsymbol{\theta}}(\mathbf{p})/K}}{Z}, \tag{5}$$

where $Z = \sum_{\mathbf{p} \in \mathcal{P}} e^{d_{\boldsymbol{\theta}}(\mathbf{p})/K}$ is a regularizer and $K$ is a temperature constant. By sampling from modeled target distribution $\pi(\mathbf{p})$ in a discrete, finite, multivariate, and non-differentiable space $\mathcal{P}$, we can obtain prompts that maximize $d_{\boldsymbol{\theta}}(\mathbf{p})$, which are likely to be memorized image trigger prompts.

---

[2]https://tineye.com/

### 3.3 Constructing MCMC by Leveraging $d_\theta$

To tackle the aforementioned challenging optimization problem, we propose to use Markov Chain Monte Carlo (MCMC) (Hastings, 1970) to sample from the target distribution $\pi(\mathbf{p})$. This method allows us to efficiently explore the discrete prompt space and find prompts likely to induce memorized images, effectively navigating $\mathcal{P}$ to identify optimal prompts. From any arbitrary distribution of sentence, $\pi_0$, Markov Chain with transition matrix $\mathbf{T}$ can be developed as follows:

$$\pi_{i+1} = \pi_i \mathbf{T}. \tag{6}$$

It is well known that Markov Chains satisfying irreducibility and aperiodicity converge to certain distribution $\pi^*$ (Robert et al., 1999), which can be formulated as $\pi_n = \pi_0 \mathbf{T}^n \to \pi^*$ independent of $\pi_0$. The transition matrix can vary depending on the algorithm used to solve the MCMC. By carefully choosing the sampling algorithm, we can ensure that the final distribution $\pi^*$ reached by the transition matrix converges to desired target distribution $\pi$ (Robert et al., 1999; Geman & Geman, 1984; Hastings, 1970). Considering the multi-dimensional nature of our parameter space, we employ the Gibbs sampling algorithm (Geman & Geman, 1984) for simplicity. Gibbs sampling is an MCMC sampling algorithm method where, at each step, only one coordinate of the multi-dimensional variable is updated to transition from the current state to the next state. Gibbs sampling algorithm has proven the convergence of the transition matrix and is known for fast convergence in multi-dimensional problems (Johnson et al., 2013; Terenin et al., 2020; Papaspiliopoulos & Roberts, 2008). We adopt random scan Gibbs sampling, which involves randomly selecting an index and updating the value at that index. This process can be expressed as the sum of $n$ transition matrices, as follows:

$$\mathbf{T} = \sum_{i=1}^{n} \frac{1}{n} \cdot \mathbf{T}_i, \tag{7}$$

$$[\mathbf{T}_i]_{\mathbf{p}^j \to \mathbf{p}^{j+1}} = \begin{cases} \pi(\mathbf{p}_i^{j+1} | \mathbf{p}_{-i}^j) & \text{if } \mathbf{p}_{-i}^j = \mathbf{p}_{-i}^{j+1} \\ 0 & \text{else,} \end{cases} \tag{8}$$

where $\mathbf{p}_{-i} = \{\mathbf{p}_1, \mathbf{p}_2, ..., \mathbf{p}_{i-1}, \mathbf{p}_{i+1}, ..., \mathbf{p}_n\}$ and $\mathbf{p}^j$ is a $j$-th state prompt. Integrating Equation 5 into the above formulas, the final transition matrix is obtained as follows:

$$[\mathbf{T}]_{\mathbf{p}^j \to \mathbf{p}^{j+1}} = \begin{cases} \frac{1}{n} \left( \dfrac{e^{d_{\boldsymbol{\theta}}(\mathcal{P}_i = \mathbf{p}_i^{j+1}, \mathcal{P}_{-i} = \mathbf{p}_{-i}^j)/K}}{\sum_{\mathbf{w} \in \mathcal{W}} e^{d_{\boldsymbol{\theta}}(\mathcal{P}_i = \mathbf{w}, \mathcal{P}_{-i} = \mathbf{p}_{-i}^j)/K}} \right) & \text{if } \mathbf{p}_{-i}^j = \mathbf{p}_{-i}^{j+1}, \\ 0 & \text{else,} \end{cases} \tag{9}$$

where detailed derivation is provided in Appendix B. Since it is impractical to compute $d_{\boldsymbol{\theta}}(\cdot)$ for all $\mathbf{w} \in \mathcal{W}$, we approximate $\mathcal{W}$ as top $Q$ samples obtained from BERT (Devlin et al., 2018). This means that the $i$-th element of the prompt $\mathbf{p}$ is masked and BERT is used to predict the word, from which the top Q samples are selected as candidate words. Additionally, this approximation can induce the generated prompts to be more semantically fluent. Mathematical derivation is complex, but the algorithm is straightforward: the process iteratively 1) selects and replace a word into [MASK] token from the sentence, 2) predicts top $Q$ words via BERT and computes proposal distribution, and 3) replaces it according to the proposal distribution. Please refer to Algorithm 1 for details.

### 3.4 Dataset Construction by Leveraging MCMC

We conduct dataset construction in two stages: 1) using a masked sentence as the prior and employing MCMC to find memorized image trigger prompts, and 2) using the memorized image trigger prompts as the prior for augmentation through MCMC.

**Using Masked Sentence as Prior.** This stage aims to discover new memorized images. The sentence is initialized with sentence of length $n$ [MASK] token, *i.e.* $\mathbf{p}_0 = \{[\text{MASK}], [\text{MASK}], ... , [\text{MASK}]\}$. We then employ Algorithm 1 initialized with $\mathbf{p}_0$ to obtain the candidate prompt. Similar to the conventional approach (Carlini et al., 2023) to extract train images, we then generate 100 images for this prompt and leverage DBSCAN (Ester et al., 1996) clustering algorithm with SSCD (Pizzi et al., 2022) to extract images

---

**Algorithm 1** Memorized Image Trigger Prompt Searching via Gibbs sampling

---

1: **Input:** Diffusion model $\boldsymbol{\theta}$, BERT model $\boldsymbol{\phi}$, initial sentence $\mathbf{p}^0$ with length $n$, iteration number $N$, number of proposal words $Q$, termination threshold $\kappa$, hyperparameter $K$, $\gamma$, $\{r_1, \ldots, r_n\}$
2: $\mathbf{p}^* \leftarrow \mathbf{p}^0$
3: **while** $d_{\boldsymbol{\theta}}(\mathbf{p}^*) < \kappa$ **do**
4:      **for** $j = 0$ to $N$ **do**
5:          Randomly select index $i \in \{1, \ldots, n\}$
6:          $\mathcal{W}_Q \leftarrow \arg \operatorname{top}_Q p_{\boldsymbol{\phi}}(\mathbf{w} \mid \mathbf{p}_{-i}^j)$
7:          $p(\mathbf{p}_i^{j+1} \mid \mathbf{p}_{-i}^j) \leftarrow \dfrac{e^{d_{\boldsymbol{\theta}}(\mathcal{P}_i = \mathbf{p}_i^{j+1}, \mathcal{P}_{-i} = \mathbf{p}_{-i}^j)/K}}{\sum_{\mathbf{w} \in \mathcal{W}_Q} e^{d_{\boldsymbol{\theta}}(\mathcal{P}_i = \mathbf{w}, \mathcal{P}_{-i} = \mathbf{p}_{-i}^j)/K}}$
8:          $\mathbf{p}_i^{j+1} \leftarrow$ Sample from $p(\mathbf{p}_i^{j+1} \mid \mathbf{p}_{-i}^j)$
9:          $\mathbf{p}^{j+1} \leftarrow (\mathbf{p}_1^j, \mathbf{p}_2^j, \ldots, \mathbf{p}_i^{j+1}, \ldots, \mathbf{p}_n^j)$
10:      **end for**
11:      $\mathbf{p}^* \leftarrow \arg \max_{\mathbf{p} \in \{\mathbf{p}^0, \mathbf{p}^1, \ldots, \mathbf{p}^n\}} d_{\boldsymbol{\theta}}(\mathbf{p})$
12: **end while**
13: **return** $\mathbf{p}^*$

---

Table 1: Comparison of the number of memorized images and trigger prompts in each dataset. Our dataset is significantly larger in terms of the number of trigger prompts across all models. Please note that images sharing the same layout, as shown in Figure 4, have been counted as a single image.

| | Stable Diffusion 1 | | Stable Diffusion 2 | | DeepFloydIF | | Realistic Vision | |
|---|---|---|---|---|---|---|---|---|
| | Trigger Prompt # | Mem. Image # | Trigger Prompt # | Mem. Image # | Trigger Prompt # | Mem. Image # | Trigger Prompt # | Mem. Image # |
| Webster (2023) | 325 | 111 | 210 | 25 | 162 | 17 | 354 | 119 |
| **MemBench** | **3000** | **151** | **1500** | **55** | **309** | **51** | **1352** | **148** |

forming at least 20 nodes. Those images are employed to Reverse Image Search API to find train image sources and human verification is conducted.

**Using Found Trigger Prompts as Prior.** This stage aims to augment memorized image trigger prompts. We leverage the prompts found in the previous stage or those provided by Webster (2023) as the prior, $\pi_0$. In this process, we employ a slightly modified algorithm to enhance diversity. Instead of running a single chain for one prompt, we run $n$ separate chains for each word position in an $n$-length sentence, treating each position as the first updating index in Gibbs sampling. This method ensures a varied exploration of the prompt space. We then save the top 100 prompts with the highest $d_{\boldsymbol{\theta}}(\cdot)$. We retained all prompts generated during the MCMC sampling process and then selected 20 augmented prompts per original prompt, considering diversity. The detailed process is provided in Appendix C.

## 4 Dataset Statistics and Efficiency of the Proposed Algorithm

### 4.1 Dataset Statistics

Table 1 presents the number of memorized images and trigger prompts obtained using the methodology described in Section 3. For both Stable Diffusion 1 and 2, the number of prompts has increased more than fivefold for each model and more than 9 times in total compared to those reported by Webster (2023). The number of memorized images included in the dataset has also increased, with Stable Diffusion 2 showing an increase of over twofold. Additionally, we provide memorized images and trigger prompts for DeepFloydIF (Shonenkov et al., 2023), which has a cascaded structure, and Realistic Vision (CivitAI, 2023), an open-source diffusion model. For these two models, we provide a larger number of memorized images and trigger prompts than Webster et al. We have also applied our algorithm to the more recent model, Stable

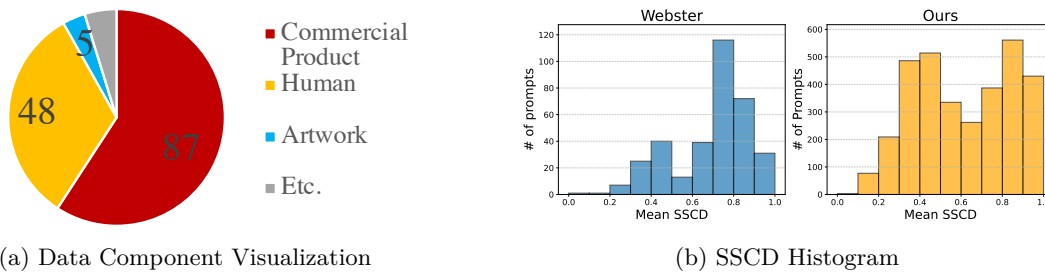

(a) Data Component Visualization                 (b) SSCD Histogram

Figure 1: (a) Components of memorized images in Stable Diffusion 1. (b) A histogram of SSCD values for the trigger prompts discovered by Webster (2023) and those in MemBench. For each trigger prompt, 10 images were generated, and the SSCD values were measured against the memorized image and then averaged. Using these averaged SSCD values, we constructed a histogram with SSCD bins for all trigger prompts from Webster (2023), and performed the same procedure for our trigger prompts. As shown, the mean SSCD values of our trigger prompts are significantly more diverse, whereas those from Webster (2023) are concentrated in the higher SSCD range.

Table 2: Comparison of the efficiency of our method and other prompt space optimization methods. Experiment was done on 1 A100 GPU. "-" denotes the failure of the valid search.

|  | Greedy Search | ZeroCap | PEZ | ConZIC | **Ours** |
|---|---|---|---|---|---|
| Hours/Memorized Image | 5.7 | - | - | 3.81 | **2.08** |

Diffusion 3 (Esser et al., 2024). Please refer to Appendix E for the results. The composition of the images included in MemBench is shown in Figure 1a, illustrating that the memorized images encompass a substantial number of commercial product images and human images. Specifically, commercial products include phone cases, furniture, clothing, and baggage; artwork includes cartoons and brand logos; and human-related images include radio albums, actors, bands, movie posters, and soccer players. Please refer to Appendix C.3 for the data component of other models.

Furthermore, we observe that the degree of memorization in the trigger prompts found by Webster (2023), $\mathbb{E}[\mathcal{M}(\mathbf{x}(\epsilon_{\boldsymbol{\theta}}, \mathbf{p}), \mathcal{D}_{train})]$, is excessively high. This presents a drawback for their direct use as benchmarks, as they consist only of prompts that are particularly difficult for mitigation methods to handle. As a result, when mitigation methods are applied with limited strength, their effectiveness may not be clearly observed, making comparisons between different approaches more challenging. This problem arises because the prompts are taken directly from the training of diffusion models. In contrast, our method searches trigger prompts through MCMC sampling in an open prompt space, allowing us to obtain a much broader range of memorization levels. We validate this by measuring the SSCD values of images generated by trigger prompts and visualizing them in a histogram, as shown in Fig. 1b.

## 4.2 Efficacy of Memorized Image Trigger Prompt Searching

In this section, we validate the efficiency of our method in discovering memorized images without access to $\mathcal{D}_{train}$. The task of finding memorized image trigger prompts without $\mathcal{D}_{train}$ is defined as follows: without any prior information, the method must automatically find trigger prompts that induce memorized images. This involves: 1) selecting candidate prompts, 2) generating 100 images for each candidate prompt, 3) applying DBSCAN (Ester et al., 1996) clustering with SSCD to get candidate images and using a Reverse Image Search API to verify those images' presence on the web. To the best of our knowledge, this task is novel, so we provide naive baselines. As the first baseline, we perform a greedy search by measuring $D_{\theta}$ for all prompts in the prompt dataset and selecting the top 200 prompts as candidate prompts. For the prompt dataset, we leveraged DiffusionDB, which contains 13M prompts collected from diffusion model users. Additionally, we provide three other baselines, all of which are algorithms that solve optimization problems in the prompt space. For two of these baselines, we adapt ZeroCap (Tewel et al., 2022) and ConZIC (Zeng et al., 2023),

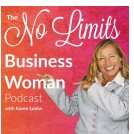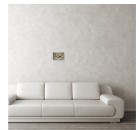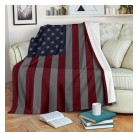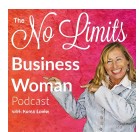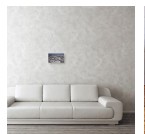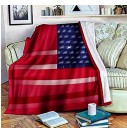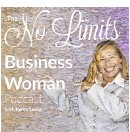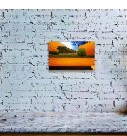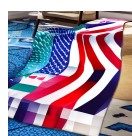

    (a) Images in LAION          (b) Stable Diffusion Generated         (c) Mitigation Applied

Figure 2: **The necessity of measuring the Aesthetic score.** Images generated with the mitigation method applied are not desirable but achieve a low SSCD while maintaining a high CLIP Score. The trigger prompts used to generate the images were: "the no limits business woman podcast", "wall view 003", and "united states throw blanket", respectively.

methods designed to maximize the CLIP Score for zero-shot image captioning, by replacing their objective function with $D_\theta$. Similarly, we also adapt PEZ (Wen et al., 2024a) by substituting its objective function with $D_\theta$ to serve as another baseline. For each of these three methods, we conducted 200 iterations and obtained prompts. For our method, we performed 200 MCMC runs with 150 iterations each, and selected the resulting prompts as candidate prompts. For more detailed implementation, please refer to Appendix G.

The results are shown in Table 2. Our method significantly outperforms other methods. The results in Table 2 demonstrate that our method significantly outperforms others. To generate 200 candidate prompts, ZeroCap and PEZ required 44 and 33 hours, respectively, on an A100 GPU but failed to identify any memorized image trigger prompts. ZeroCap's sequential prediction hindered the prompt being optimized to have higher $D_\theta$ values than general prompts. For PEZ, we observed that the prompts were optimized to produce images with a specific color (e.g., sunflower fields, grassy fields), and the prompts themselves were very unnatural. ConZIC identified 6 memorized images in 24 hours but struggled with local minima and lacked diversity in its optimization process, resulting in lower efficiency compared to our method.

Comparing with baselines (Carlini et al., 2023; Webster, 2023) that leverage LAION itself is challenging, as the dataset is no longer available and the elements for reimplementation are omitted in the corresponding papers. However, as mentioned in Section 2, their memory-inefficient and computationally intensive methods provided only a few memorized images and trigger prompts.

## 5 MemBench: Metrics, Scenarios and Reference Performance

**Metrics.** We present rigorous metrics for correctly evaluating mitigation methods, which include similarity score, Text-Image alignment score, and quality score. Following previous works, we adopt **SSCD** (Pizzi et al., 2022) as the similarity score and measure max SSCD between a generated image using trigger prompt and memorized images. In detail, if a prompt $\mathbf{p}^*$ triggers images $\{\mathbf{x}_1^*, ..., \mathbf{x}_k^*\}$ included in $\mathcal{D}_{train}$, we measure $\max_{\mathbf{x} \in \{\mathbf{x}_1^*, ..., \mathbf{x}_k^*\}} SSCD(\mathbf{x}(\mathbf{p}^*, \boldsymbol{\epsilon}_\theta), \mathbf{x})$. Secondly, we adopt **CLIP Score** (Hessel et al., 2021) to measure Text-Image alignment between prompt and generated images. Lastly, We adopt an **Aesthetic Score** (Schuhmann et al., 2022) as the image quality score. While previous works (Wen et al., 2024b; Somepalli et al., 2023b) did not assess image quality scores, we observed issues shown in Figure 2. When memorization mitigation methods are applied, we observe that image quality degrades, the rich context generated by the diffusion model is destroyed, or distorted images are formed. While Ren et al. (2024) measured FID, they reported that FID decreases when mitigation methods are applied. They attribute this phenomenon to the mitigation method preventing memorized images from being generated, thereby increasing the diversity of generated images. As a result, compared to FID which focuses on diversity, Aesthetic Score offers a more straightforward way to evaluate individual image quality and highlight image quality degradation issues.

**Scenarios.** To ensure that memorization mitigation methods can be generally applied to diffusion models, we provide two scenarios: the memorized image trigger prompt scenario and the general prompt scenario. First, the memorized image trigger prompt scenario evaluates whether mitigation methods can effectively prevent the generation of memorized images. This scenario uses the memorized image trigger prompts we identified in Section 3. We generate 10 images for each trigger prompt and measure the Top-1 SSCD and the mean values of the Top-3 SSCD. We also measure the proportion of images with SSCD exceeding 0.5.

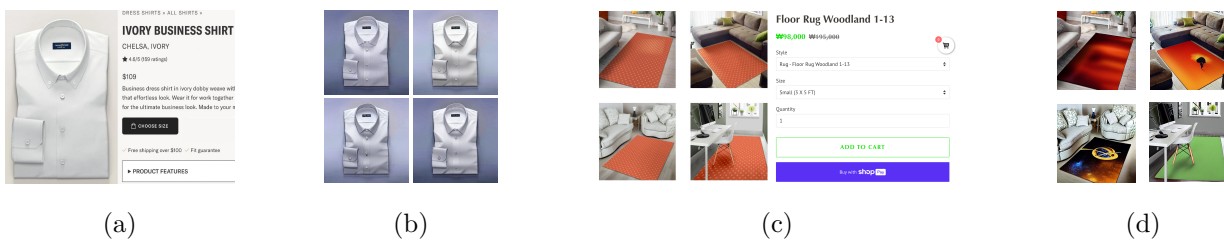

|  (a)  |  (b)  |  (c)  |  (d)  |

Figure 3: Examples of memorized images found using the Reverse Image Search API. (a), (c) Shirt/rug currently sold commercially, (b), (d) four images generated by Stable Diffusioon

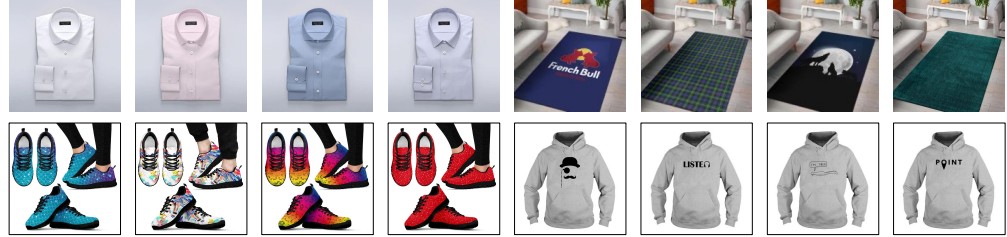

Figure 4: Results of images found by leveraging Reverse Image Search API to the images generated from trigger prompts. While prior works (Carlini et al., 2023; Webster, 2023) have attributed memorization in diffusion models to image repetition, this does not explain why memorization still occurs in LAION-5B, where image deduplication has been applied. We suggest that layout duplication shown in the figure may account for such remaining cases of memorization.

For CLIP Score and Aesthetic Score, we calculate the average value across all generated images. Second, the general prompt scenario ensures that the performance of the diffusion model does not degrade when using prompts other than trigger prompts. We leverage the COCO (Lin et al., 2014) validation set as general prompts. In this scenario, images are generated once per prompt, and the average CLIP Score and Aesthetic Score are measured.

**Reference Performance.**  We propose a reference performance for interpreting the performance of mitigation methods. An effective mitigation method should be able to reduce SSCD while maintaining CLIP Score. However, although SSCD is a metric designed to compare the structural similarity of images for copy detection tasks, it inevitably includes semantic meaning due to the self-supervised nature of the trained neural network. On the other hand, the semantic meaning of the trigger prompt should still be reflected in the generated image to maintain CLIP Score even when a mitigation method is applied. Therefore, it is uncertain how much the SSCD between memorized images and generated images can be reduced while maintaining the CLIP Score between trigger prompts and generated images. In this regard, we provide a reference performance to indicate how much SSCD can be reduced while maintaining a high CLIP Score. We assume querying images with trigger prompts via the Google Image API[3] as a strong proxy model for the generative model and provide the reference performance. Please refer to Appendix D for details.

## 6    Deeper Analysis into Image Memorization

In this section, we provide a deeper analysis of the memorized images and trigger prompts in MemBench. Firstly, we have found that **Stable Diffusion regenerates commercial products currently on sale**. While prior works (Carlini et al., 2023; Somepalli et al., 2023a) have suggested the possibility that diffusion models could memorize commercial images, we are the first to confirm this. Unlike the previous studies that used image retrieval from LAION to find memorized images (Carlini et al., 2023; Webster, 2023), we leverage a Reverse Image Search API to find those, which enable us this verification. As shown in Figure 3.b, Stable

---

[3]https://developers.google.com/custom-search

Table 3: Performance evaluation of image memorization mitigation methods in MemBench. $n$, $l$, and $c$ are hyperparameters that control the strength of their corresponding mitigation methods. Please refer to Appendix G.3 for the details of hyper-parameters.

| | Hyper-params | MemBench | | | | | | COCO | | |
| | | Top-1 SSCD ↓ | Top-3 SSCD ↓ | SSCD > 0.5 ↓ | CLIP ↑ | Aesth. ↑ | Aesth. std. ↓ | CLIP ↑ | Aesth. ↑ | Aesth. std. ↓ |
|---|---|---|---|---|---|---|---|---|---|---|
| Base | | 0.641 | 0.605 | 0.451 | 0.273 | 5.25 | 0.43 | 0.321 | 5.37 | 0.36 |
| **Reference Performance** | | **0.088** | - | - | **0.310** | - | - | - | - | - |
| RNA(Somepalli et al., 2023b) | n = 1 | 0.479 | 0.425 | 0.241 | 0.270 | 5.18 | 0.53 | 0.314 | 5.34 | 0.36 |
| | n = 2 | 0.389 | 0.338 | 0.165 | 0.270 | 5.14 | 0.55 | 0.310 | 5.33 | 0.37 |
| | n = 3 | 0.329 | 0.280 | 0.121 | 0.267 | 5.13 | 0.56 | 0.307 | 5.30 | 0.37 |
| | n = 4 | 0.287 | 0.239 | 0.089 | 0.264 | 5.10 | 0.58 | 0.304 | 5.29 | 0.39 |
| | n = 5 | 0.254 | 0.213 | 0.074 | 0.262 | 5.08 | 0.59 | 0.302 | 5.28 | 0.39 |
| | n = 6 | 0.228 | 0.189 | 0.055 | 0.258 | 5.06 | 0.59 | 0.298 | 5.24 | 0.38 |
| RTA (Somepalli et al., 2023b) | n = 1 | 0.497 | 0.446 | 0.265 | 0.269 | 5.20 | 0.52 | 0.316 | 5.34 | - |
| | n = 2 | 0.397 | 0.347 | 0.175 | 0.268 | 5.19 | 0.53 | 0.314 | 5.32 | 0.36 |
| | n = 3 | 0.330 | 0.285 | 0.129 | 0.266 | 5.17 | 0.54 | 0.310 | 5.29 | 0.36 |
| | n = 4 | 0.282 | 0.240 | 0.094 | 0.264 | 5.15 | 0.55 | 0.306 | 5.27 | 0.37 |
| | n = 5 | 0.257 | 0.217 | 0.080 | 0.262 | 5.14 | 0.53 | 0.302 | 5.26 | 0.37 |
| | n = 6 | 0.228 | 0.190 | 0.056 | 0.258 | 5.10 | 0.56 | 0.299 | 5.27 | 0.38 |
| Wen et al. (2024b) | l = 7 | 0.410 | 0.346 | 0.134 | 0.270 | 5.16 | 0.54 | 0.321 | 5.37 | 0.36 |
| | l = 6 | 0.355 | 0.289 | 0.089 | 0.270 | 5.15 | 0.55 | 0.321 | 5.37 | 0.36 |
| | l = 5 | 0.312 | 0.246 | 0.059 | 0.269 | 5.14 | 0.56 | 0.321 | 5.37 | 0.36 |
| | l = 4 | 0.259 | 0.199 | 0.035 | 0.268 | 5.13 | 0.57 | 0.321 | 5.37 | 0.36 |
| | l = 3 | 0.181 | 0.139 | 0.015 | 0.264 | 5.11 | 0.59 | 0.321 | 5.37 | 0.36 |
| | l = 2 | 0.096 | 0.075 | 0.001 | 0.242 | 4.97 | 0.64 | 0.321 | 5.37 | 0.36 |
| Ren et al. (2024) | c = 1.0 | 0.289 | 0.247 | 0.083 | 0.263 | 5.17 | 0.57 | 0.316 | 5.33 | 0.38 |
| | c = 1.1 | 0.283 | 0.239 | 0.071 | 0.260 | 5.17 | 0.57 | 0.313 | 5.31 | 0.38 |
| | c = 1.2 | 0.278 | 0.232 | 0.058 | 0.257 | 5.15 | 0.58 | 0.309 | 5.28 | 0.39 |
| | c = 1.3 | 0.275 | 0.227 | 0.050 | 0.254 | 5.14 | 0.58 | 0.304 | 5.26 | 0.39 |

Diffusion replicates images of commercially available shirts when given a specific prompt. Figure 3.d further illustrates the replication of layouts; for a commercially sold carpet, all layouts have been reproduced.

Secondly, we explore the cause of image memorization in Stable Diffusion 2, trained on LAION-5B, whose duplicates are removed. Previous works (Somepalli et al., 2023b; Gu et al., 2023) suggested that image memorization issues arise from duplicate images in the training data. Webster et al. (2023) confirmed that the LAION-2B dataset contains many duplicate images likely to be memorized. However, Stable Diffusion 2 still exhibits image memorization issues while reduced. We hypothesize that this memorization arises due to layout duplication. Figure 4 shows the images found by Reverse Image Search API that are memorized by Stable Diffusion. We found that there are often over 100 images on the web with the same layout but different color structures. LAION-5B underwent deduplication based on URLs[4], but this process may not have removed these images. These layout memorizations are also obviously subject to copyright, posing potential social issues. Additional examples are provided in Appendix H.

## 7 Evaluation of Image Memorization Mitigation Methods

In this section, we evaluate image memorization mitigation methods on our MemBench in Stable Diffusion 1. For results of Stable Diffusion 2, please refer to Appendix F.2.

**Baselines.** We use Stable Diffusion 1.4 as the base model. The image memorization mitigation methods evaluated include: 1) RTA (Somepalli et al., 2023b), which applies random token insertion to the prompt, 2) RNA (Somepalli et al., 2023b), which inserts a random number between $[0, 10^6]$ into the prompt, 3) method proposed by Wen et al. (2024b) that applies adversarial attacks to text embeddings, and 4) method proposed by Ren et al. (2024) that rescales cross-attention. Image generation is performed using the DDIM (Song et al., 2021a) Scheduler with a guidance scale of 7.5 and 50 inference steps.

**Results.** We present the experimental results in Table 3. As shown in Table 3, for all methods, lowering the SSCD significantly reduces both the CLIP Score and the Aesthetic Score. This indicates a degradation in text-image alignment and image quality. In particular, upon examining images with low Aesthetic Scores, we observe that issues in Figure 2 occur across all methods. Moreover, when hyper-parameters are set as high

---

[4]https://laion.ai/blog/laion-5b/

values for mitigation methods, it leads not only to a lower Aesthetic Score but also to a much larger standard deviation. This indicates that diffusion model outputs become unreliable. Please refer to Appendix F.1 for the FID values. As reported by Wen et al. (2024b), all methods exhibit a trade-off between SSCD and CLIP Score. Regarding the reference performance obtained via API search, it can be observed that the SSCD can be reduced to 0.088 while maintaining a high CLIP Score. Due to the inherent limitations of the Stable Diffusion baseline model, the CLIP Score cannot exceed 0.273 when mitigation methods are applied. However, mitigation methods should aim to reduce the Top-1 SSCD to around 0.088 while maintaining at least this level of CLIP Score.

To provide a more detailed analysis of each method, we observe that the approach proposed by Wen et al. achieves the best performance in the trade-off between SSCD and CLIP Score. However, to reduce the proportion of images with SSCD exceeding 0.5—indicative of image memorization—to nearly zero, their method still requires a reduction in CLIP Score by 0.025. Given the scale of the CLIP Score, this drop suggests that the generated images may be only marginally related to the given prompts. Moreover, a significant decrease in the Aesthetic Score is also observed. On the other hand, the method proposed by Wen et al. has an additional advantage: it does not result in any performance drop in the general prompt scenario on the COCO dataset, making it the most suitable option for practical applications as of now.

The most recent method proposed by Ren et al. (2024) shows a considerable reduction in the CLIP Score. Even at the lowest hyper-parameter setting ($c = 1.0$), the reductions in both CLIP Score and Aesthetic Score are substantial, limiting its general applicability to diffusion models. The most basic approaches, RNA and RTA, show a decrease in CLIP Score by 0.015 at the hyper-parameter setting ($n = 6$) that lowers the proportion of images with SSCD exceeding 0.5 to 0.05. This is expected, given the nature of these methods: both attempt to prevent image memorization by adding irrelevant tokens to the prompts. As a result, RNA and RTA are unreliable for application to diffusion models.

## 8 Conclusion

We have presented MemBench, the first benchmark for evaluating image memorization mitigation methods in diffusion models. MemBench includes various memorized image trigger prompts, appropriate metrics, and a practical scenario to ensure that mitigation methods can be effectively applied in practice. We have provided the reference performance that mitigation methods should aim to achieve. Through MemBench, we have confirmed that existing image memorization mitigation methods are still insufficient for application to diffusion models in practical scenarios. The lack of a benchmark may have previously hindered the research of effective mitigation methods. However, we believe that our benchmark will facilitate significant advancements in this field.

**Limitations and Future Work.** Another contribution of our work is providing an algorithm for efficiently searching memorized image trigger prompts based on MCMC. Our approach is faster than other searching algorithms we have tried, yet it does not exhibit exceptionally high speed. Consequently, due to time constraints, we were unable to provide a larger number of memorized images. However, our method allows for the continuous search of more memorized images and their corresponding trigger prompts, and we plan to update the dataset regularly. Additionally, we aim to enhance the efficiency of our memorized image trigger prompt searching algorithm in the future.

## Acknowledgments

The work was supported by GPU resources provided by Artificial Intelligence Industry Cluster (AICA). T.-H. Oh is partially supported by Institute of Information & communications Technology Planning & Evaluation (IITP) grant funded by the Korea government(MSIT) (No. RS-2024-00457882, National AI Research Lab Project; No.RS-2022-II220124, Development of Artificial Intelligence Technology for Self-Improving Competency-Aware Learning Capabilities). M. Sung was partially supported by the NRF grant (RS-2023-00209723) and IITP grants (RS-2022-II220594, RS-2023-00227592, RS-2024-00399817), all funded by the Korean government (MSIT).

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

# Appendix

## Contents

## A   Broader Impact Statement

Our work introduces a technique for extracting the training data of diffusion models. This could potentially harm the rights of model owners or image copyright holders. Therefore, it is crucial to handle this technique with caution to avoid any infringement issues. For more details, please refer to Appendix I.

## B   Detailed Derivation of Transition Matrix

In this section, we provide a detailed derivation of the transition matrix that was omitted in Section 3. To recap, the transition matrix of the random scan Gibbs sampler for sampling the target distribution $\pi$ is defined as follows:

$$\mathbf{T} = \sum_{i=1}^{n} \frac{1}{n} \cdot \mathbf{T}_i, \tag{10}$$

$$[\mathbf{T}_i]_{\mathbf{p}^j \to \mathbf{p}^{j+1}} = \begin{cases} \pi(\mathbf{p}_i^{j+1}|\mathbf{p}_{-i}^j) & \text{if } \mathbf{p}_{-i}^j = \mathbf{p}_{-i}^{j+1} \\ 0 & \text{else,} \end{cases} \tag{11}$$

where $n$ is the total length of sentence, $\mathbf{p}_{-i} = \{\mathbf{p}_1, \mathbf{p}_2, ..., \mathbf{p}_{i-1}, \mathbf{p}_{i+1}, ..., \mathbf{p}_n\}$ and $\mathbf{p}^j$ is a $j$-th state prompt. We proceed to derive the conditional probability distribution of the target distribution $\pi(\mathbf{p}) = \frac{e^{d_{\boldsymbol{\theta}}(\mathbf{p})/K}}{Z}$:

$$\pi(\mathbf{p}_i^{j+1} \mid \mathbf{p}_{-i}^j) = \frac{\pi(\mathcal{P}_i = \mathbf{p}_i^{j+1}, \mathcal{P}_{-i} = \mathbf{p}_{-i}^j)}{\pi(\mathbf{p}_{-i}^j)} \tag{12}$$

$$= \frac{\pi(\mathcal{P}_i = \mathbf{p}_i^{j+1}, \mathcal{P}_{-i} = \mathbf{p}_{-i}^j)}{\sum_{\mathbf{w} \in \mathcal{W}} \pi(\mathcal{P}_i = \mathbf{w}, \mathcal{P}_{-i} = \mathbf{p}_{-i}^j)} \tag{13}$$

$$= \frac{e^{d_{\boldsymbol{\theta}}(\mathcal{P}_i = \mathbf{p}_i^{j+1}, \mathcal{P}_{-i} = \mathbf{p}_{-i}^j)/K}}{\sum_{\mathbf{w} \in \mathcal{W}} e^{d_{\boldsymbol{\theta}}(\mathcal{P}_i = \mathbf{w}, \mathcal{P}_{-i} = \mathbf{p}_{-i}^j)/K}} \tag{14}$$

By substituting Equation 14 into Equation 11, we ultimately derive the transition matrix as defined earlier in Equation 9.

$$[\mathbf{T}]_{\mathbf{p}^j \to \mathbf{p}^{j+1}} = \begin{cases} \frac{1}{n} \cdot \left( \frac{e^{d_{\boldsymbol{\theta}}(\mathcal{P}_i = \mathbf{p}_i^{j+1}, \mathcal{P}_{-i} = \mathbf{p}_{-i}^j)/K}}{\sum_{\mathbf{w} \in \mathcal{W}} e^{d_{\boldsymbol{\theta}}(\mathcal{P}_i = \mathbf{w}, \mathcal{P}_{-i} = \mathbf{p}_{-i}^j)/K}} \right) & if \ \mathbf{p}_{-i}^j = \mathbf{p}_{-i}^{j+1}, \\ 0 & else. \end{cases} \tag{15}$$

## C   Data Construction Details

In this section, we provide a detailed explanation of the data construction process described in Section 3.4. We explain 1) how memorized image trigger prompts and corresponding memorized images for Stable Diffusion 1 and 2 in Table 1 were found, 2) implementation details of the data augmentation algorithm using MCMC, and 3) its efficiency.

For Stable Diffusion 1 and Realistic Vision, we initialized sentences with $n$-length mask tokens and implemented Algorithm 1 to find new memorized image trigger prompts and corresponding memorized images (please refer to Section 3.4 **"Using Masked Sentence as Prior"**). We then perform the MCMC process with $\mathbf{p}_0$ initialized by trigger prompts to perform the augmentation (please refer to Section 3.4 **"Using Found Trigger Prompts as Prior"**). For Stable Diffusion 2 and DeepFloydIF, the process of finding trigger prompts using masked sentences was omitted. This was due to two reasons: firstly, the prediction accuracy

---

**Algorithm 2** Memorized Image Trigger Prompt Augmentation via Gibbs Sampling

---

1: **Input:** Diffusion model $\boldsymbol{\theta}$, BERT model $\boldsymbol{\phi}$, initial sentence $\mathbf{p}^0$ with length $n$, iteration number $N$, number of proposal words $Q$, termination threshold $\kappa$, early stop counter threshold $s$, hyperparameter $K$.
2: Initialize early stop counter $c \leftarrow 0$
3: Initialize prompt bank $\mathcal{B} \leftarrow \{\mathbf{p}^0\}$
4: **for** $k = 0$ to $n$ **do**
5:     **for** $j = 0$ to $N$ **do**
6:         **if** $j = 0$ **then**
7:             $i \leftarrow k$
8:         **else**
9:             Randomly select index $i \in \{1, \ldots, n\}$
10:         **end if**
11:         $\mathcal{W}_Q \leftarrow \arg\text{top}_Q \ p_{\boldsymbol{\phi}}(\mathbf{w} \mid \mathbf{p}^j_{-i})$
12:         $p(\mathbf{p}^{j+1}_i \mid \mathbf{p}^j_{-i}) \leftarrow \dfrac{e^{d_{\boldsymbol{\theta}}(\mathcal{P}_i=\mathbf{p}^{j+1}_i, \mathcal{P}_{-i}=\mathbf{p}^j_{-i})/K}}{\sum_{\mathbf{w} \in \mathcal{W}_Q} e^{d_{\boldsymbol{\theta}}(\mathcal{P}_i=\mathbf{w}, \mathcal{P}_{-i}=\mathbf{p}^j_{-i})/K}}$
13:         $\mathbf{p}^{j+1}_i \leftarrow$ Sample from $p(\mathbf{p}^{j+1}_i \mid \mathbf{p}^j_{-i})$
14:         $\mathbf{p}^{j+1} \leftarrow (\mathbf{p}^j_1, \mathbf{p}^j_2, \ldots, \mathbf{p}^{j+1}_i, \ldots, \mathbf{p}^j_n)$
15:         Add $\{(\mathbf{p}^j_1, \mathbf{p}^j_2, \ldots, \mathbf{p}^j_{i-1}, \mathbf{w}, \mathbf{p}^j_{i+1}, \ldots, \mathbf{p}^j_n) \mid \forall \mathbf{w} \in \mathcal{W}_Q\}$ to $\mathcal{B}$
16:         **if** $d_{\boldsymbol{\theta}}(\mathbf{p}^{j+1}) < \kappa$ **then**
17:             $c \leftarrow c + 1$
18:         **else**
19:             $c \leftarrow 0$
20:         **end if**
21:         **if** $c > s$ **then**
22:             **break**
23:         **end if**
24:     **end for**
25: **end for**
26: **return** $\mathcal{B}$

---

**Algorithm 3** Diversity Sampling

---

1: **Input:** Text encoder $\boldsymbol{\phi}$, augmented prompts $\mathcal{B}$, return prompts number $N$
2: Randomly select $\mathbf{p}^* \in \mathcal{B}$
3: Initialize return prompt list $\mathcal{R} \leftarrow \{\mathbf{p}^*\}$
4: **while** $|\mathcal{R}| < N$ **do**
5:     $\mathbf{p}^* \leftarrow \arg\min_{\mathbf{p} \in \mathcal{B}} \max_{\mathbf{p}_r \in \mathcal{R}} \dfrac{\boldsymbol{\phi}(\mathbf{p}) \cdot \boldsymbol{\phi}(\mathbf{p}_r)}{\|\boldsymbol{\phi}(\mathbf{p})\| \|\boldsymbol{\phi}(\mathbf{p}_r)\|}$
6:     $\mathcal{B} \leftarrow \mathcal{B} \setminus \{\mathbf{p}^*\}$
7:     $\mathcal{R} \leftarrow \mathcal{R} \cup \{\mathbf{p}^*\}$
8: **end while**
9: **return** $\mathcal{R}$

---

of $d_{\boldsymbol{\theta}}$ for memorized image trigger prompts is lower for these models. Secondly, as Stable Diffusion 2 is trained on the deduplicated LAION-5B and LAION-A, the memorized image trigger prompts are sparser, making optimization from a masked sentence initialization difficult. Therefore, for Stable Diffusion 2 and DeepFloydIF, only the trigger prompt augmentation algorithm was leveraged. The prompts were initialized in two ways before undergoing the data augmentation process: 1) using trigger prompts found from Stable Diffusion 1, and 2) using trigger prompts provided by Webster (2023). We further elaborate on the data augmentation process below.

## C.1   Data Augmentation Leveraging MCMC

Trigger prompt augmentation was carried out using a different approach from trigger prompt searching (Algorithm 1). The process of generating candidate trigger prompts through prompt augmentation is detailed in Algorithm 2. We initialized $\mathbf{p}_0$ with the trigger prompt itself and then performed MCMC. As explained in Section 3.4, in trigger prompt augmentation, we run $n$ separate chains for each word position in an $n$-length sentence, treating each position as the first updating index in Gibbs sampling. During the MCMC process, all prompts with calculated $d_{\boldsymbol{\theta}}$ values were stored in the prompt bank. Additionally, we adopted an early stop counter. The prompts returned by Algorithm 2 tend to have low diversity due to the nature of Gibbs Sampling. Therefore, Algorithm 3 is applied to all returned prompts to create a smaller, more diverse subset of prompts. Afterward, these prompts undergo an image generation process, followed by human verification, before being added to the dataset.

## C.2   Data Augmentation Performance

We present an evaluation of Algorithm 2, our proposed method for augmenting memorized image trigger prompts. To assess the effectiveness of Algorithm 2, we examine whether the prompts generated during the algorithm's execution indeed trigger memorized images. Although Algorithm 2 is designed to return only the top $T$ candidate trigger prompts, for this experiment, we investigate all the prompts generated during the execution of Algorithm 2 to measure its performance. Given the extensive time required to verify all candidate trigger prompts, we present a toy experiment focusing on a specific prompt: "The no limits business woman podcast," which generates an image identical to Figure 5.a. For the experiment, we set the hyperparameters of Algorithm 2 as follows: $K = 1.5$, $N = 50$, $Q = 200$, $\kappa = 3$, and $s = 3$. The experiment was conducted using a single A100 GPU.

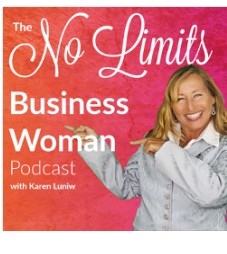 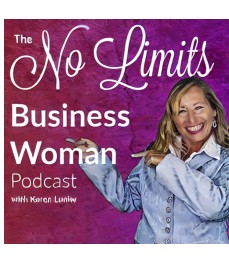

(a)                     (b)

Figure 5: Memorized image utilized for toy experiment. Each image refers to (a) train data image in Stable Diffusion, (b) generated image using Stable Diffusion. The SSCD between (a) and (b) is measured to be 0.707.

For those prompts generated during Algorithm 2, we filtered only prompts that show $d_{\boldsymbol{\theta}}(\mathbf{p}) > 5$ and generated 10 images for each. Then we measure the Top-1 SSCD (Pizzi et al., 2022) with the image in Figure 5.a. We found that there were 4217 unique prompts with a Top-1 SSCD exceeding 0.7, indicating that they replicate train data image (as seen in Figure 5). Algorithm 2 took 7 minutes on an A100 GPU, producing 4217 augmented trigger prompts within this time frame. In addition, we categorized these prompts based on the number of words changed from the original prompt. Specifically, there were 753 trigger prompts with one word changed, 1923 with two words changed, 1352 with three words changed, 179 with four words changed, and 10 with five words changed. Interestingly, even with the modification of five out of the six words in the sentence, the altered prompts can still effectively induce memorized images. This demonstrates our method's efficiency in generating a large number of augmented trigger prompts in a short period.

## C.3   Detailed Data Component of MemBench

In addition to the component analysis of memorized images in Stable Diffusion 1 presented in Sec. 4.1, we extend this analysis to Stable Diffusion 2, Realistic Vision, and DeepFloydIF. The results are shown in Figure 6. Notably, memorized images related to Humans disappear in both Stable Diffusion 2 and DeepFloydIF. We attribute this to the fact that both models were trained on datasets with image duplication removed. As discussed in Sec. 6, image memorization in such cases is likely due to layout duplication. Since human-related images are less likely to involve layout-level duplication, they are less prone to being memorized under these conditions.

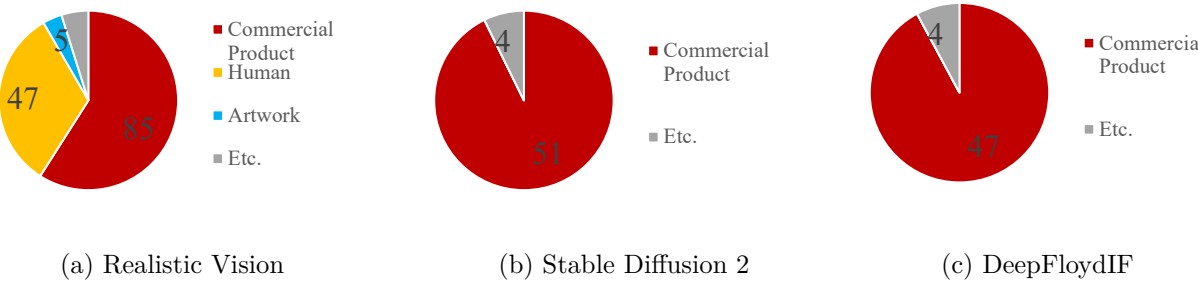

(a) Realistic Vision            (b) Stable Diffusion 2            (c) DeepFloydIF

Figure 6: Components of memorized images in Realistic Vision, Stable Diffusion 2, and DeepFloydIF.

## D  Reference Performance Based on Image Search API

In Section 5, we posited that the Google Image Search API[5] serves as a strong proxy model for the generative model and presented the measured reference performance. We provide the details of this approach in this section. Before explaining our use of the Google Image Search API, it is important to reiterate our goal in presenting reference performance: to determine the extent to which the SSCD (Pizzi et al., 2022) between the memorized image and the generated image can be minimized while maintaining the semantic content of the trigger prompt. To address this question, we utilized the Google Image Search API to measure reference performance as follows: 1) Query 100 images using the memorized image trigger prompt via the API. 2) Measure the CLIP Score (Hessel et al., 2021) between the 100 images and the trigger prompt. 3) Retain only the image with the Top-1 CLIP Score. 4) Measure the SSCD between this retained image and the memorized image triggered by the prompt in Stable Diffusion. 5) Repeat steps 1-4 for all memorized image trigger prompts in MemBench. After completing these steps, we reported the average Top-1 CLIP Score and the average SSCD of images with the Top-1 CLIP Score in Section 7. Our findings show that the SSCD can be reduced to 0.200 while maintaining a CLIP Score of 0.329. This indicates that the minimum achievable SSCD with maintaining CLIP Score is 0.210. Therefore, we should strive to develop mitigation methods that achieve this or better. Please note that we did not measure Aesthetic Score (Schuhmann et al., 2022) and evaluate the reference performance in COCO (Lin et al., 2014) settings, since comparing them with the mitigation method is not meaningful.

## E  Extension to Stable Diffusion 3

We applied our algorithm to Stable Diffusion 3. However, as the training data for Stable Diffusion 3 is publicly unknown (no information is available), we were unable to perform the verification process, Reverse Image Search API. Thus, the searched images and prompts for Stable Diffusion 3 cannot serve as a memorization benchmark. To be used as a memorization benchmark, two critical steps are essential: 1) Candidate Trigger Prompt Search Step and 2) Verification Step, where we confirm whether the repeated images actually exist in the training data, thereby verifying that they are indeed memorized images, as noted in Section 3.2. Without the verification step, we are not sure whether the searched images and prompts are memorized. Nevertheless, in Figure 7, we present the trigger prompts and duplicated images identified by our MCMC algorithm for Stable Diffusion 3. Although we cannot verify them due to the aforementioned limitations, we believe they represent strong candidates for memorized images.

## F  Evaluation of Image Memorization Mitigation Method on MemBench

### F.1  FID of Mitigation Methods Measured on MemBench

In this section, we present the FID values measured on MemBench when applying mitigation methods to Stable Diffusion 1. As shown in Table 4, FID values increase when mitigation methods are applied. This

---

[5]https://developers.google.com/custom-search

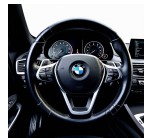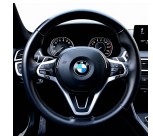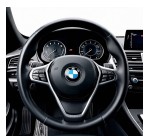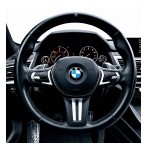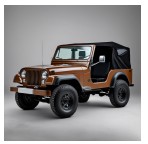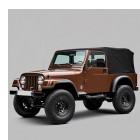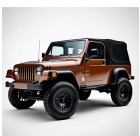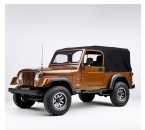

Vibrant ebony bmw rising inside steering wheel.  Walnut size lan exclusive jeep operating customs.

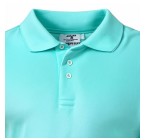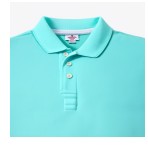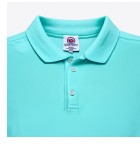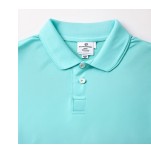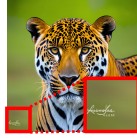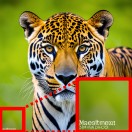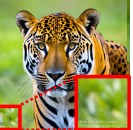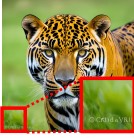

Label above aqua pale translucent polo shirt.  Fiery eyes mature female leopard aka tigers.

Figure 7: Trigger prompt searched by our MCMC algorithm and generated images with corresponding prompt. The repeated and very similar images strongly suggest the occurrence of memorization. Notably, the last images show a tendency to repeat specific text at the bottom left. This is a common feature of memorized images generated by diffusion models, where text, URLs, or similar elements from the source image are replicated. This strongly suggests that these repeated images are indeed memorized images.

Table 4: FID scores of mitigation methods measured on MemBench.

|  | Base | RTA (Somepalli et al., 2023b) | RNA (Somepalli et al., 2023b) | Wen et al. (2024b) | Ren et al. (2024) |
|---|---|---|---|---|---|
| FID ↓ | 116.07 | 75.33 | 85.32 | 64.21 | 86.79 |

aligns with the findings reported by Ren et al. (2024), as FID also captures diversity. Since the Stable Diffusion model generates identical images for trigger prompts, the generated images exhibit low diversity, leading to higher FID values. In contrast, when mitigation methods are applied, memorized images are not generated, resulting in increased diversity and consequently lower FID values. Therefore, FID does not effectively measure image quality but rather measures diversity. Image quality should instead be assessed using the Aesthetic Score we propose.

## F.2 Evaluation of Image Memorization Mitigation Method on Other Diffusion Models

In this section, we evaluate image memorization mitigation methods on our MemBench in other diffusion models. We present the experimental results of Stable Diffusion 2 in Table 5. When each memorization mitigation method is applied, although SSCD (Pizzi et al., 2022) is reduced, there is a drop in both CLIP Score (Hessel et al., 2021) and Aesthetic Score (Schuhmann et al., 2022). Additionally, compared to the reference performance provided by the Google Image Search API, the performance of these methods is insufficient. The method proposed by Wen et al. (2024b) shows less capability in reducing SSCD while maintaining the CLIP Score compared to RNA (Somepalli et al., 2023b) and RTA (Somepalli et al., 2023b) in the memorized image trigger prompt scenario. However, in the practical scenario of the COCO validation set, its performance remains equivalent to the base Stable Diffusion 2. We present the experimental results of DeepFloydIF in Table 6

## G   Details of Experiments

### G.1   Implementation Details of Baselines in Memorized Image Trigger Prompt Searching Experiment

In this section, we provide implementation details of other baselines that we have tried to search the memorized image trigger prompts, presented in Section 4.2. All three algorithms (Wen et al., 2024a; Zeng et al., 2023; Tewel et al., 2022) that we provide are originally intended to solve various optimization problems in the text space, using specific objective functions. For our experiments to search for memorized image trigger prompts, we replaced each method's objective function with $D_\theta$.

Table 5: Performance evaluation of image memorization mitigation methods in MemoBench for Stable Diffusion 2.

| | | MemBench | | | | | COCO | |
|---|---|---|---|---|---|---|---|---|
| | | Top-1 SSCD ↓ | Top-3 SSCD ↓ | SSCD > 0.5 ↓ | CLIP ↑ | Aesthetic ↑ | CLIP ↑ | Aesthetic ↑ |
| Base | | 0.629 | 0.593 | 0.448 | 0.281 | 5.41 | 0.333 | 5.35 |
| **Reference Performance** | (API search) | **0.207** | - | - | **0.301** | - | - | - |
| RNA(Somepalli et al., 2023b) | n = 1 | 0.568 | 0.525 | 0.349 | 0.278 | 5.34 | 0.328 | 5.35 |
| | n = 2 | 0.539 | 0.491 | 0.289 | 0.276 | 5.30 | 0.326 | 5.35 |
| | n = 3 | 0.501 | 0.446 | 0.224 | 0.273 | 5.24 | 0.324 | 5.35 |
| | n = 4 | 0.453 | 0.395 | 0.161 | 0.271 | 5.21 | 0.322 | 5.35 |
| | n = 5 | 0.424 | 0.368 | 0.130 | 0.270 | 5.19 | 0.320 | 5.34 |
| RTA (Somepalli et al., 2023b) | n = 1 | 0.590 | 0.549 | 0.365 | 0.276 | 5.35 | 0.332 | 5.34 |
| | n = 2 | 0.562 | 0.515 | 0.317 | 0.275 | 5.31 | 0.330 | 5.31 |
| | n = 3 | 0.529 | 0.475 | 0.261 | 0.272 | 5.26 | 0.329 | 5.30 |
| | n = 4 | 0.479 | 0.428 | 0.211 | 0.272 | 5.21 | 0.325 | 5.27 |
| | n = 5 | 0.452 | 0.393 | 0.167 | 0.271 | 5.17 | 0.324 | 5.25 |
| Wen et al. (2024b) | l = 70 | 0.577 | 0.535 | 0.311 | 0.273 | 5.30 | 0.333 | 5.35 |
| | l = 60 | 0.553 | 0.502 | 0.251 | 0.269 | 5.26 | 0.333 | 5.35 |
| | l = 50 | 0.501 | 0.423 | 0.154 | 0.263 | 5.19 | 0.333 | 5.35 |
| | l = 40 | 0.398 | 0.322 | 0.065 | 0.253 | 5.15 | 0.333 | 5.35 |
| Ren et al. (2024) | c = 1.0 | 0.592 | 0.556 | 0.419 | 0.273 | 5.40 | 0.331 | 5.36 |
| | c = 1.1 | 0.586 | 0.548 | 0.391 | 0.270 | 5.39 | 0.326 | 5.33 |
| | c = 1.2 | 0.580 | 0.539 | 0.349 | 0.267 | 5.37 | 0.320 | 5.30 |
| | c = 1.3 | 0.574 | 0.529 | 0.295 | 0.262 | 5.34 | 0.313 | 5.27 |

| | Top-1 SSCD ↓ | Top-3 SSCD ↓ | SSCD > 0.5 ↓ | CLIP ↑ | Aesthetic ↑ |
|---|---|---|---|---|---|
| Base | 0.619 | 0.576 | 0.467 | 0.279 | 5.05 |
| Wen et al. (2024b) | 0.614 | 0.572 | 0.467 | 0.266 | 5.06 |
| RNA(Somepalli et al., 2023b) | 0.497 | 0.446 | 0.280 | 0.270 | 4.96 |
| RTA(Somepalli et al., 2023b) | 0.476 | 0.426 | 0.240 | 0.276 | 4.94 |

Table 6: Evaluation of Memorization Mitigation Method on DeepFloydIF. The hyperparameters used are as follows: RTA and RNA were set with $n = 4$, Wen et al. (2024b) with $l = 1$. The method proposed by Wen et al. (2024b) relies on applying adversarial attacks to the model, but it seems to be less effective due to the structural characteristics of DeepFloydIF, which utilizes a cascaded diffusion architecture.

**ZeroCap (Tewel et al., 2022).** ZeroCap is an optimization method developed for zero-shot image captioning tasks. This method leverages a pre-trained CLIP (Radford et al., 2021) to measure the CLIP similarity between an image and the current caption, and manipulate the prompt to maximize this CLIP Score, searching the best caption that describes the image. ZeroCap predicts the next word using a large language model (LLM) and sequentially adds tokens to the prompt in a manner that maximizes CLIP similarity. Additionally, a Context Cache is introduced for gradient descent, where the Context Cache is a set of key-value pairs derived when the current prompt is embedded into the LLM. The optimization function is consistute of 1) CLIP similarity loss between the image and the prompt, and 2) the cross-entropy (CE) loss between the distribution of the predicted token of the original Context Cache and that of the updated Context Cache. ZeroCap performs gradient descent on the optimization function to update the Context Cache five times, after which the token predicted by this Context Cache is designated as the next token to continue the sentence. Furthermore, beam search is utilized in this process. In our experiments, we replaced the CLIP similarity loss with $D_{\boldsymbol{\theta}}$ and implemented the algorithm accordingly. All hyper-parameters were set to match those in the original paper.

To generate 200 candidate prompts using ZeroCap, it took approximately 44 hours on an A100 GPU, yet not a single memorized image trigger prompt was found. While $D_{\boldsymbol{\theta}}$ values were higher compared to those of general prompts (captions from COCO validation set), they were still lower than the values for actual trigger prompts. This suggests an inherent issue with ZeroCap's sequential prediction method.

**PEZ (Wen et al., 2024a).**   The PEZ algorithm is an optimization technique designed to find prompts that will induce a diffusion model to generate a specific desired image. The algorithm operates in two main steps for each iteration: 1) perform gradient descent on the prompt in the continuous space with respect to the diffusion model's CLIP model, and 2) project the updated prompt back into the discrete space of the CLIP's embedding space. In our adaptation of this algorithm, we utilized $D_{\boldsymbol{\theta}}(\mathbf{p})$ as the objective function for calculating the gradient. All hyper-parameters were set to match those in the original paper.

To generate 200 candidate prompts using PEZ, it took approximately 33 hours on an A100 GPU. However, similar to ZeroCap, not a single memorized image trigger prompt was found. Although the $D_{\boldsymbol{\theta}}$ values were comparable to those of actual trigger prompts, memorized images were not discovered. Upon inspection, we observed that the prompts were optimized to produce images with a specific color (e.g., sunflower fields, grassy fields), and the prompts themselves were very unnatural. This suggests that the optimization process did not result in the desired memorized image trigger prompts.

**ConZIC (Zeng et al., 2023).**   ConZIC, like ZeroCap, is a technique designed to optimize the CLIP Score for zero-shot image captioning tasks. Similar to our approach, ConZIC selects a single word within the sentence, predicts the word using BERT, and then replace it with the word which shows the highest value of objective function. The objective function here is a sum of the CLIP similarity and the conditional probability distribution from BERT. In our experiments, we substituted the CLIP similarity with $D_{\boldsymbol{\theta}}$ as the objective function.

To generate 200 candidate prompts using ConZIC, it took approximately 24 hours on an A100 GPU. Unlike the other methods, ConZIC successfully identified 6 memorized images. However, ConZIC's optimization process is designed to consistently update the prompt to maximize the objective function, which tends to result in getting stuck in local minima and the lack of diversity. These lead to less efficiency compared to our method.

### G.2   Hyper-Parameters in Image Memorization Mitigation Methods

In Section 7, we evaluated the performance of image memorization mitigation methods on MemBench and presented the results in Table 3. However, due to space constraints, we omitted the explanations of various hyper-parameters in the table. Here, we provide a detailed explanation of these hyper-parameters. Firstly, RTA (Somepalli et al., 2023b) and RNA (Somepalli et al., 2023b) are methods that insert random words or numbers into the prompt. The parameter $n$ in the table indicates the number of words or numbers inserted. The method proposed by Wen et al. (2024b) involves updating the prompt embedding to minimize $D_{\boldsymbol{\theta}}$. Here, the threshold for lowering $D_{\boldsymbol{\theta}}(\mathbf{p})$, denoted as the early stopping loss $l$, becomes a hyper-parameter, *i.e.* the prompt $\mathbf{p}$ is updated until $D_{\boldsymbol{\theta}}(\mathbf{p}) < l$. All other hyper-parameters followed the settings in the original paper: an Adam optimizer with a learning rate of 0.05 and a maximum of 10 steps was used for training. Ren et al. (2024) provides a method that inversely amplifies the attention score for the beginning token by adjusting the input logits of the softmax operator in the cross-attention. To be precise, let the original input logits be denoted as $\mathbf{s} = (s_1, s_2, \ldots, s_N)$, where $s_i$ is the logit of the $i$-th token. The re-scaled logit vector $\mathbf{s}'$ is:

$$\mathbf{s}' = (Cs_1, s_2, \ldots, s_{N-S}, -\infty, ..., -\infty). \tag{16}$$

Here, the scale factor $C$ for the beginning token $s_1$ becomes a hyper-parameter. Additionally, as shown in Table 3, when $C = 1$, the performance differs significantly from the base Stable Diffusion. This is because the input logits for the summary token are all replaced with negative infinity.

### G.3   Hyper-Parameters in Memorized Image Trigger Prompt Searching Leveraging MCMC

In Table 7, we present the hyper-parameters used in our algorithm for finding memorized image trigger prompts via MCMC.

Table 7: Hyper-parameters leveraged in memorized image trigger prompt searching using our algorithm. Here, $n$ represents the sentence length, $N$ is the iteration number, $Q$ denotes the number of proposal words, $K$ stands for the temperature, $\kappa$ is the termination threshold, $s$ is the early stop counter threshold, and $T$ is the number of return candidate prompts.

| Model | Method | $n$ | $N$ | $Q$ | $K$ | $\kappa$ | $s$ | $T$ |
|---|---|---|---|---|---|---|---|---|
| Stable Diffusion 1 | Algorithm 1 | 8 | 150 | 200 | 0.1 | 5 | - | - |
| | Algorithm 2 | - | 20 | 200 | 1.5 | 3 | 3 | 100 |
| Stable Diffusion 2 | Algorithm 2 | - | 20 | 200 | 5.0 | 50 | 3 | 100 |

# H    Additional Examples of Memorized Images

In this section, we present the trigger prompts identified by our algorithm along with the generated images from Stable Diffusion using these prompts. Additionally, for each image, we provide the corresponding images presumed to be from the training data, identified using the Reverse Image Search API. The layout repetition of the generated images and those found through the API strongly indicate that Stable Diffusion has memorized the training data. Moreover, we have confirmed that the majority of these images are currently available for commercial sale. We leveraged DDIM (Song et al., 2021a) scheduler to generate images.

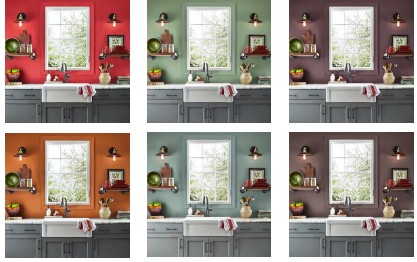
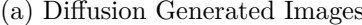
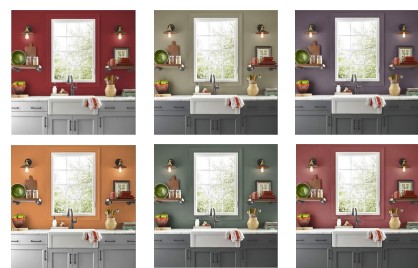

(a) Diffusion Generated Images                    (b) API Searched Images

Figure 8: Examples of memorized images found using the Reverse Image Search API. The prompt used for image generation is "Cozy kitchen painted".


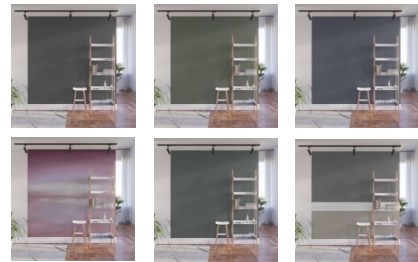

(a) Diffusion Generated Images                    (b) API Searched Images

Figure 9: Examples of memorized images found using the Reverse Image Search API. The prompt used for image generation is "Grey standard wall mural".

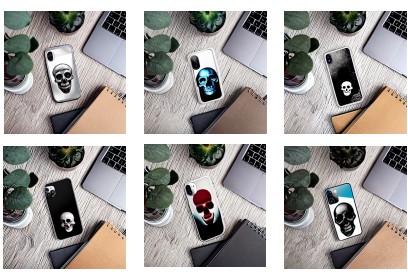

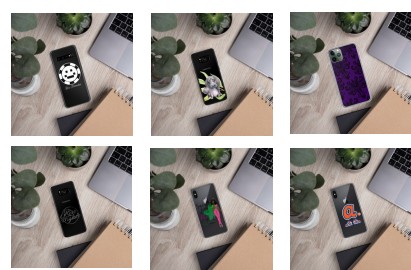

(a) Diffusion Generated Images

(b) API Searched Images

Figure 10: Examples of memorized images found using the Reverse Image Search API. The prompt used for image generation is "Iphone case covered with skull".

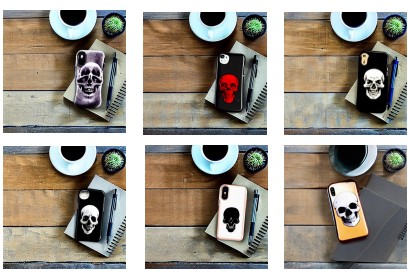

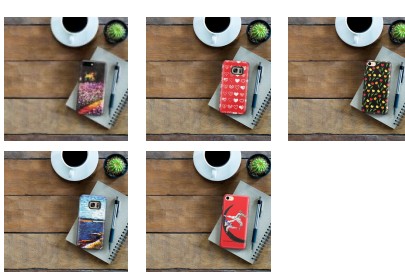

(a) Diffusion Generated Images

(b) API Searched Images

Figure 11: Examples of memorized images found using the Reverse Image Search API. The prompt used for image generation is "Iphone case covered with skull".

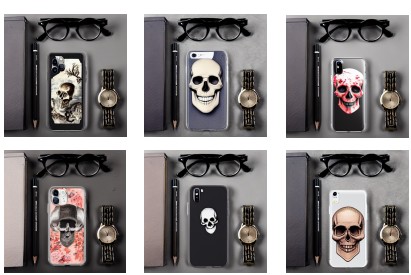

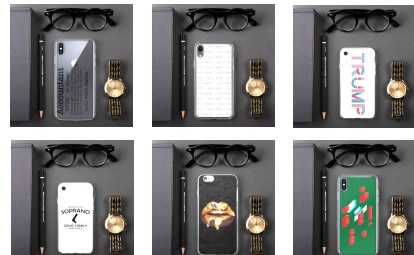

(a) Diffusion Generated Images

(b) API Searched Images

Figure 12: Examples of memorized images found using the Reverse Image Search API. The prompt used for image generation is "Iphone case covered with skull".

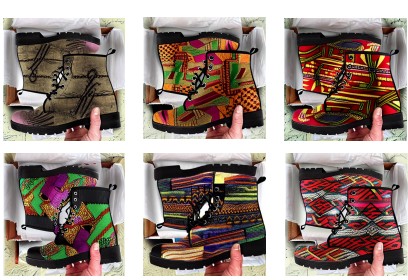



(a) Diffusion Generated Images

(b) API Searched Images

Figure 13: Examples of memorized images found using the Reverse Image Search API. The prompt used for image generation is "Knit line Africa American quilt house lace boots".

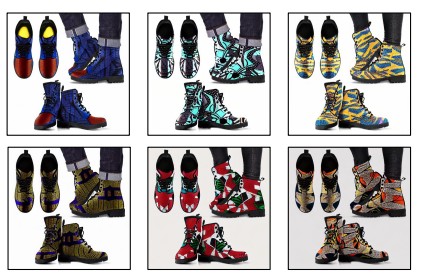



(a) Diffusion Generated Images

(b) API Searched Images

Figure 14: Examples of memorized images found using the Reverse Image Search API. The prompt used for image generation is "Knit line Africa American quilt house lace boots".

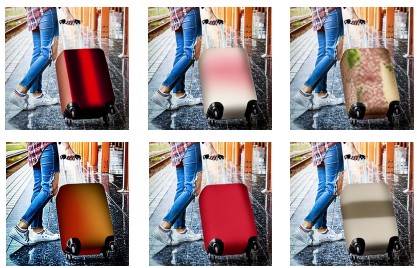

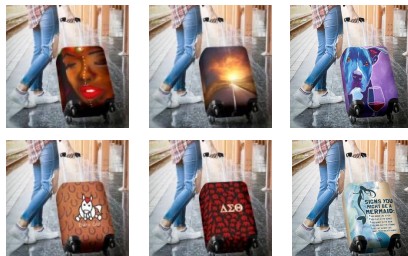

(a) Diffusion Generated Images

(b) API Searched Images

Figure 15: Examples of memorized images found using the Reverse Image Search API. The prompt used for image generation is "Travel luggage cover".



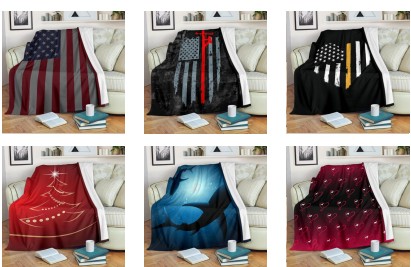

(a) Diffusion Generated Images

(b) API Searched Images

Figure 16: Examples of memorized images found using the Reverse Image Search API. The prompt used for image generation is "United states throw blanket".

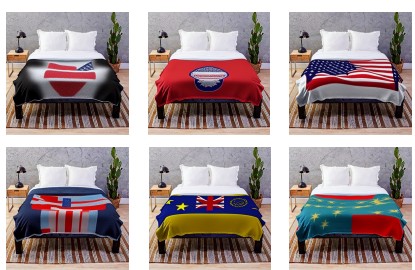

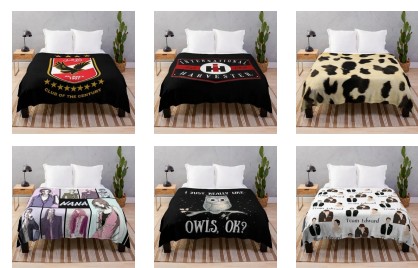

(a) Diffusion Generated Images

(b) API Searched Images

Figure 17: Examples of memorized images found using the Reverse Image Search API. The prompt used for image generation is "United states throw blanket".

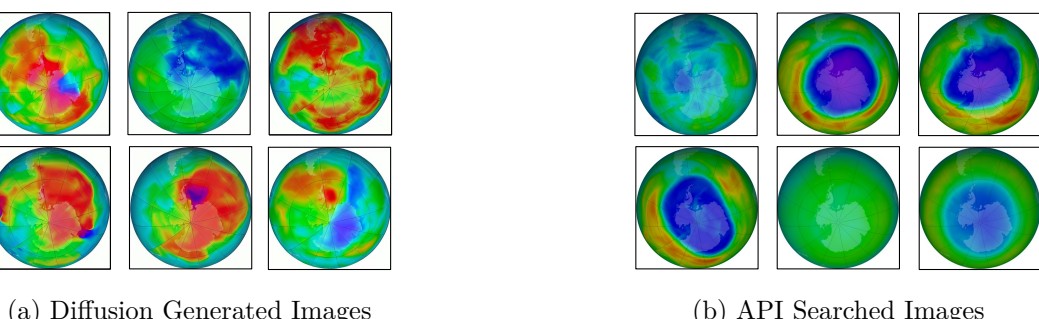

(a) Diffusion Generated Images         (b) API Searched Images

Figure 18: Examples of memorized images found using the Reverse Image Search API. The prompt used for image generation is "Uranus center as ozone temperature map".

# I Datasheet

## I.1 Motivation

Q1 **For what purpose was the dataset created?** Was there a specific task in mind? Was there a specific gap that needed to be filled? Please provide a description.

- MemBench is a benchmark designed for evaluating memorization mitigation methods in diffusion models. Recently, many diffusion models have been highlighted for their issues with image memorization, prompting the development of various memorization mitigation methods. However, due to the absence of a benchmark to properly evaluate these methods, their effectiveness has not been adequately assessed. To address this, we developed MemBench, which includes a large number of memorized image trigger prompts and appropriate metrics for evaluation.

Q2 **Who created the dataset (e.g., which team, research group) and on behalf of which entity (e.g., company, institution, organization)?**

- Chunsan Hong, Tae-Hyun Oh, and Minhyuk Sung from KAIST created the dataset.

Q3 **Who funded the creation of the dataset?** If there is an associated grant, please provide the name of the grantor and the grant name and number.

- Please refer to Acknowledgement section (Sec. 8).

Q4 **Any other comments?**

- No.

## I.2 Composition

Q5 **What do the instances that comprise the dataset represent (e.g., documents, photos, people, countries)?** *Are there multiple types of instances (e.g., movies, users, and ratings; people and interactions between them; nodes and edges)? Please provide a description.*

- It includes links to the images memorized by Text-to-Image diffusion models and the prompts that trigger these images.

Q6 **How many instances are there in total (of each type, if appropriate)?**

- Please refer to Section 1

Q7 **Does the dataset contain all possible instances or is it a sample (not necessarily random) of instances from a larger set?** *If the dataset is a sample, then what is the larger set? Is the sample representative of the larger set (e.g., geographic coverage)? If so, please describe how this representativeness was validated/verified. If it is not representative of the larger set, please describe why not (e.g., to cover a more diverse range of instances, because instances were withheld or unavailable).*

- It will be a sample of all existing trigger prompts that induce memorized images in Stable Diffusion. However, to the best of our knowledge, we have secured the largest number of trigger prompts, and we plan to add more in the future.

Q8 **What data does each instance consist of?** *"Raw" data (e.g., unprocessed text or images) or features? In either case, please provide a description.*

- Input trigger prompt and URLs of memorized images triggered by the corresponding prompt.

Q9 **Is there a label or target associated with each instance?** *If so, please provide a description.*

- No.

Q10 **Is any information missing from individual instances?** *If so, please provide a description, explaining why this information is missing (e.g., because it was unavailable). This does not include intentionally removed information, but might include, e.g., redacted text.*

- No.

Q11 **Are relationships between individual instances made explicit (e.g., users' movie ratings, social network links)?** *If so, please describe how these relationships are made explicit.*

- Yes, in our benchmark, relationships between individual instances are made explicit. For example, each trigger prompt in our dataset is explicitly linked to the memorized image it induces. This is done by including pairs of trigger prompts and their corresponding memorized images, clearly showing the relationship between them. Additionally, each image in the benchmark is linked to the specific Text-to-Image diffusion model that memorized it, providing a clear mapping of model-instance relationships.

Q12 **Are there recommended data splits (e.g., training, development/validation, testing)?** *If so, please provide a description of these splits, explaining the rationale behind them.*

- No.

Q13 **Are there any errors, sources of noise, or redundancies in the dataset?** *If so, please provide a description.*

- No.

Q14 **Is the dataset self-contained, or does it link to or otherwise rely on external resources (e.g., websites, tweets, other datasets)?** *If it links to or relies on external resources, a) are there guarantees that they will exist, and remain constant, over time; b) are there official archival versions of the complete dataset (i.e., including the external resources as they existed at the time the dataset was created); c) are there any restrictions (e.g., licenses, fees) associated with any of the external resources that might apply to a future user? Please provide descriptions of all external resources and any restrictions associated with them, as well as links or other access points, as appropriate.*

- The dataset includes the URLs of the memorized images.
- Regarding (a), we provide multiple URLs for each memorized image to ensure the dataset's longevity, even if one hosting source goes down.
- Regarding (b), since the memorized images are not copyrighted by us, we cannot provide the images directly.
- Regarding (c), these images should be used solely for evaluating the effectiveness of mitigation methods and should not be used for commercial training or distribution.

Q15 **Does the dataset contain data that might be considered confidential (e.g., data that is protected by legal privilege or by doctor–patient confidentiality, data that includes the content of individuals' non-public communications)?** *If so, please provide a description.*

- No.

Q16 **Does the dataset contain data that, if viewed directly, might be offensive, insulting, threatening, or might otherwise cause anxiety?** *If so, please describe why.*

- No.

Q17 **Does the dataset relate to people?** *If not, you may skip the remaining questions in this section.*

- No.

Q18 **Does the dataset identify any subpopulations (e.g., by age, gender)?**

- No.

Q19 **Is it possible to identify individuals (i.e., one or more natural persons), either directly or indirectly (i.e., in combination with other data) from the dataset?** *If so, please describe how.*

- The memorized images include faces of celebrities, such as Emma Watson.

Q20 **Does the dataset contain data that might be considered sensitive in any way (e.g., data that reveals racial or ethnic origins, sexual orientations, religious beliefs, political opinions or union memberships, or locations; financial or health data; biometric or genetic data; forms of government identification, such as social security numbers; criminal history)?** *If so, please provide a description.*

- No.

Q21 **Any other comments?**

- Although the memorized images we discovered do not contain offensive or confidential elements, they do include images of currently sold products and faces of celebrities. For instance, there are images of Emma Watson. Therefore, these images should be used solely for evaluating the effectiveness of memorization mitigation methods.

## I.3 Collection Process

Q22 **How was the data associated with each instance acquired?** *Was the data directly observable (e.g., raw text, movie ratings), reported by subjects (e.g., survey responses), or indirectly inferred/derived from other data (e.g., part-of-speech tags, model-based guesses for age or language)? If data was reported by subjects or indirectly inferred/derived from other data, was the data validated/verified? If so, please describe how.*

- Our method discovers memorized images without any prior information by leveraging BERT models, diffusion models, and the Reverse Image Search API. For more details, please refer to Section 3.

Q23 **What mechanisms or procedures were used to collect the data (e.g., hardware apparatus or sensor, manual human curation, software program, software API)?** *How were these mechanisms or procedures validated?*

- Our method employs the Reverse Image Search API[6] and Google Image Search API[7] to discover memorized images. For more details, please refer to Section 3, 5.

Q24 **If the dataset is a sample from a larger set, what was the sampling strategy (e.g., deterministic, probabilistic with specific sampling probabilities)?**

- No.

Q25 **Who was involved in the data collection process (e.g., students, crowdworkers, contractors) and how were they compensated (e.g., how much were crowdworkers paid)?**

- None. The process was automated.

Q26 **Over what timeframe was the data collected? Does this timeframe match the creation timeframe of the data associated with the instances (e.g., recent crawl of old news articles)?** *If not, please describe the timeframe in which the data associated with the instances was created.*

- The data was collected from April 2024 to May 2024.

---

[6]https://tineye.com/
[7]https://developers.google.com/custom-search

Q27 **Were any ethical review processes conducted (e.g., by an institutional review board)?** *If so, please provide a description of these review processes, including the outcomes, as well as a link or other access point to any supporting documentation.*

- No.

Q28 **Does the dataset relate to people?** *If not, you may skip the remaining questions in this section.*

- People may appear in the memorized images.

Q29 **Did you collect the data from the individuals in question directly, or obtain it via third parties or other sources (e.g., websites)?**

- Our method employs the Reverse Image Search API and Google Image Search API to discover memorized images. For more details, please refer to Section 3, 5.

Q30 **Were the individuals in question notified about the data collection?** *If so, please describe (or show with screenshots or other information) how notice was provided, and provide a link or other access point to, or otherwise reproduce, the exact language of the notification itself.*

- Our automated memorized image trigger prompt searching algorithm did not involve any participation of individuals.

Q31 **Did the individuals in question consent to the collection and use of their data?** *If so, please describe (or show with screenshots or other information) how consent was requested and provided, and provide a link or other access point to, or otherwise reproduce, the exact language to which the individuals consented.*

- Our automated memorized image trigger prompt searching algorithm did not involve any participation of individuals.

Q32 **If consent was obtained, were the consenting individuals provided with a mechanism to revoke their consent in the future or for certain uses?** *If so, please provide a description, as well as a link or other access point to the mechanism (if appropriate).*

- Our automated memorized image trigger prompt searching algorithm did not involve any participation of individuals.

Q33 **Has an analysis of the potential impact of the dataset and its use on data subjects (e.g., a data protection impact analysis) been conducted?** *If so, please provide a description of this analysis, including the outcomes, as well as a link or other access point to any supporting documentation.*

- We discuss the limitation of our current work in Section 8, and we plan to further investigate and analyze the impact of our benchmark in future work.

Q34 **Any other comments?**

- No.

### I.4 Preprocessing, Cleaning, and/or Labeling

Q35 **Was any preprocessing/cleaning/labeling of the data done (e.g., discretization or bucketing, tokenization, part-of-speech tagging, SIFT feature extraction, removal of instances, processing of missing values)?** *If so, please provide a description. If not, you may skip the remainder of the questions in this section.*

- No.

Q36 **Was the "raw" data saved in addition to the preprocessed/cleaned/labeled data (e.g., to support unanticipated future uses)?** *If so, please provide a link or other access point to the "raw" data.*

- N/A.

Q37 **Is the software used to preprocess/clean/label the instances available?** *If so, please provide a link or other access point.*

- N/A.

Q38 **Any other comments?**

- No.

## I.5 Uses

Q39 **Has the dataset been used for any tasks already?** *If so, please provide a description.*

- Not yet. MemBench is a new benchmark.

Q40 **Is there a repository that links to any or all papers or systems that use the dataset?** *If so, please provide a link or other access point.*

- Not yet. We plan to provide links to works that use our benchmark.

Q41 **What (other) tasks could the dataset be used for?**

- Image memorization mitigation in diffusion models.

Q42 **Is there anything about the composition of the dataset or the way it was collected and preprocessed/cleaned/labeled that might impact future uses?** *For example, is there anything that a future user might need to know to avoid uses that could result in unfair treatment of individuals or groups (e.g., stereotyping, quality of service issues) or other undesirable harms (e.g., financial harms, legal risks) If so, please provide a description. Is there anything a future user could do to mitigate these undesirable harms?*

- No.

Q43 **Are there tasks for which the dataset should not be used?** *If so, please provide a description.*

- The images we provide must not be used for training generative models. Since these images include faces of celebrities and currently sold products, they should never be used or distributed for the training of generative models.

Q44 **Any other comments?**

- No.

## I.6 Distribution and License

Q45 **Will the dataset be distributed to third parties outside of the entity (e.g., company, institution, organization) on behalf of which the dataset was created?** *If so, please provide a description.*

- Yes, this benchmark will be open-source.

Q46 **How will the dataset be distributed (e.g., tarball on website, API, GitHub)?** *Does the dataset have a digital object identifier (DOI)?*

- The code and datasets are available at https://github.com/chunsanHong/MemBench_code

Q47 **When will the dataset be distributed?**

- It has already been distributed through the above link.

Q48 **Will the dataset be distributed under a copyright or other intellectual property (IP) license, and/or under applicable terms of use (ToU)?** *If so, please describe this license and/or ToU, and provide a link or other access point to, or otherwise reproduce, any relevant licensing terms or ToU, as well as any fees associated with these restrictions.*

- The dataset will be distributed under the Creative Commons Attribution 4.0 International (CC BY 4.0) license for the URLs and trigger prompts, not for the images themselves, as the images themselves are not owned by us. We will provide terms of use document specifying that the dataset is intended solely for research and evaluation of memorization mitigation methods and should not be used for training generative models. The GitHub repository, where the benchmark will be distributed, will contain the code licensed under the MIT License. The terms of use and licensing information will be accessible via the GitHub repository when it becomes available.

Q49 **Have any third parties imposed IP-based or other restrictions on the data associated with the instances?** *If so, please describe these restrictions, and provide a link or other access point to, or otherwise reproduce, any relevant licensing terms, as well as any fees associated with these restrictions.*

- Yes, third parties own the images referenced by the URLs in our dataset. These images include those of celebrities and currently sold products, which are protected under their respective intellectual property rights. The URLs provided are for reference purposes only and must not be used for training or commercial distribution. Any use of the images must comply with the respective third-party terms and conditions. There are no fees associated with these restrictions, but users must respect the IP rights of the original content owners.

Q50 **Do any export controls or other regulatory restrictions apply to the dataset or to individual instances?** *If so, please describe these restrictions, and provide a link or other access point to, or otherwise reproduce, any supporting documentation.*

- No.

Q51 **Any other comments?**

- No.

## I.7 Maintenance

Q52 **Who will be supporting/hosting/maintaining the dataset?**

- Chunsan Hong, the first author, will maintain the dataset.

Q53 **How can the owner/curator/manager of the dataset be contacted (e.g., email address)?**

- Through the GitHub discussions that will be opened soon.
- Through the email of the author.

Q54 **Is there an erratum?** *If so, please provide a link or other access point.*

- No.

Q55 **Will the dataset be updated (e.g., to correct labeling errors, add new instances, delete instances)?** *If so, please describe how often, by whom, and how updates will be communicated to users (e.g., mailing list, GitHub)?*

- MemBench will be updated. We plan to search for more memorized image trigger prompts and corresponding memorized images using our continuous algorithm.

Q56 **If the dataset relates to people, are there applicable limits on the retention of the data associated with the instances (e.g., were individuals in question told that their data would be retained for a fixed period of time and then deleted)?** *If so, please describe these limits and explain how they will be enforced.*

- N/A.

Q57 **Will older versions of the dataset continue to be supported/hosted/maintained?** *If so, please describe how. If not, please describe how its obsolescence will be communicated to users.*

- We will host other versions.

Q58 **If others want to extend/augment/build on/contribute to the dataset, is there a mechanism for them to do so?** *If so, please provide a description. Will these contributions be validated/verified? If so, please describe how. If not, why not? Is there a process for communicating/distributing these contributions to other users? If so, please provide a description.*

- Through the email of the author.

Q59 **Any other comments?**

- No.

