# OpenReview forum: "MemBench: Memorized Image Trigger Prompt Dataset for Diffusion Models"
_TMLR — Accepted by TMLR_

### Review · Reviewer_LXqR · 2025-03-10

**Summary Of Contributions:**

The main contribution of this paper is a benchmark (MemBench) for the memorized image trigger prompt and their retrieved image from the web for several T2I diffusion models; the direct user of this particular benchmark would be the modeling community on memorization mitigation methods. Specifically, this benchmark is larger than the prior works and does not rely on direct access to the LAION dataset. For the evaluation side, the authors propose incorporating several additional metrics for assessing the existing mitigation methods, pointing out that current evaluation often overlooks the side effects of memorization mitigation of sacrificing the general generation quality, measured by Aesthetic Score.

**Audience:**

Yes

**Broader Impact Concerns:**

I reviewed the Datasheet in the appendix and have a few concerns.

First, the quality assessment of the proposed benchmark appears to be largely missing. The most apparent advantage over prior work seems to be its larger dataset size, but without a proper evaluation of prompt and image quality, it is difficult to assess its overall reliability. I strongly recommend conducting a comprehensive human evaluation, following properly documented IRB regulations, to ensure the benchmark’s quality and integrity.

Additionally, the authors provide URLs to memorized images from the web, raising potential copyright and legal risks. It is unclear whether this practice is legally permissible or how such risks might impact future research built upon this benchmark. The editor may want to consider consulting an external ethical reviewer to further assess these concerns.

**Claims And Evidence:**

Yes

**Requested Changes:**

**In addition** to the major weaknesses pointed out in the previous section, I have several more detailed comments regarding the requested changes.

- Figure 1 should be improved, the current version only has a pie chart with even no unit on the scores. Also, as I have previously mentioned, the statistics breakdown w.r.t. different base models should be included and analyzed.

- In Figure 2, the corresponding text prompts used to generate those figures should also be added.

- The layout and the content from Figure 3 and Figure 4 are somewhat similar, they are expected to be much more informative and should be merged.

- In Table 3, why use different notations for API search? What do n, l, and c mean, and why do they have different numeracy accuracies?

- The mathematical notation in Sec 3.2 has some inconsistencies. For instance, *...to find prompts yielding high $\mathcal{D}_\theta$*, should be D$_\theta$ instead. Also use $\mathcal{D}$ for ensemble and D for measure is already confusing.

- Can you please also provide the diversity of the prompt in the proposed benchmark?

**Strengths And Weaknesses:**

**S1**: The proposed MemBench tackles a task that might be of interest to some of the researchers in the modeling community, which is the memorization mitigation problem in diffusion models.

**S2**: It makes sense to consider the general quality of the generated images when mitigating the memorization issue, which makes the proposed point of including an additional Aesthetic Score reasonable.

**S3**: The proposed prompt searching does not require direct access to the original LAION dataset and leverages MCMC sampling instead, which is intuitively reasonable.

**W1**: Some of the presentation (e.g., figures and mathematical notations) and writing need further improvement. Please see my detailed comments in the Requested Changes section.

**W2**: The positioning between this work and the previous works (e.g., Webster 2023) is rather vague and unclear. While I hear the authors mention that the existing search methods usually require access to the training dataset LAION, it does not directly impact the target user of this particular benchmark. In other words, as a potential modeling researcher working on mitigation methods, direct access to LAISON in the benchmark construction process does not provide additional value anyway. That being said, I think the authors need to come up with stronger practical values and advantages, such as the quality of the benchmark itself as the key strength to convince the community. And this quality comparison (which should be independent from the downstream task performance) is currently missing, as there are no human evaluations to verify the actual quality of the proposed MemBench, the only dimension emphasized is the size and the search algorithm used to construct the benchmark.

**W3**: Analysis of the proposed benchmark is insufficient. For instance, I believe a model-wise analysis of the dataset statistics would be helpful in Sec. 4.1. The proposed benchmark includes trigger prompts for different diffusion T2I models, and the model-wise breakdown would provide valuable insights on how the memorization is established in different T2I variants.

**W4**: I read Sec. 6 for the deeper analysis into image memorization. While I appreciate the authors' efforts to a deeper understanding of this memorization issue, unfortunately found it does not appear to be very fundamental. Memorization is directly related to diffusion training dynamics embedded in the formulation, several theoretical works [a] have explicitly pointed out the connection between the collapse phase transition in learning dynamics and image memorization, the authors are advised to look into this line of theoretical works to better understand the underlying theoretical mechanisms of memorizations.

**W5**: While I agree that we should look into other dimensions, such as the general quality, when developing mitigation methods, I am not fully convinced of the rationale for replacing FID with Aesthetic Score, can you please also provide the FID scores for the benchmarked methods in Table 3, as comparison to the reported Aesthetic Score.

[a] Dynamical regimes of diffusion models, Nature Communications 2024

---

> ### Author Response · Authors · 2025-04-12
>
> We appreciate the reviewer’s detailed comments. Due to the large number of suggestions, we have grouped similar points together, which may have altered their order. We kindly ask the reviewer to confirm that all points have been addressed.
>
>
> **W1. Presentation and writing need improvement.**
>
>
> We appreciate the valuable suggestions. We have addressed all the points raised and incorporated the changes into the revised version. A brief summary is as follows:
> - **Suggestion 1: Fig. 1 (pie chart) should be improved.)**  We have analyzed the pie chart for different base models and included those results in Appendix B.3. However, we are unsure about the exact meaning of “including a score in the pie chart.” If the reviewer clarify this, we will address it accordingly.
> - **Suggestion 2: In Fig. 2, the corresponding trigger prompts should be added.)** We have added the prompts directly in Fig. 2.
> - **Suggestion 3: Fig. 3 and Fig. 4 should be merged.)**   We believe that Fig. 3 and Fig. 4 serve distinct purposes, and merging them may cause confusion for readers. Therefore, we prefer to keep them separate. To clarify their roles:   1) Fig. 3 demonstrates that certain memorized images are actually for sale on commercial websites.   2) Fig. 4 provides a hypothesis regarding another cause of diffusion memorization—layout duplication.  Specifically, while prior works [C1,2,3] have primarily attributed memorization in diffusion models to image repetition, this does not explain why memorization still occurs in LAION-5B, where image deduplication has been applied. Fig. 4 aims to suggest that layout duplication may account for such remaining cases of memorization.   To further emphasize this distinction, we have added a detailed explanation to the caption of Fig. 4 in the revised version.
> - **Suggestion 4: In Table 3, why use different notations for API search? (What do *n*, *l*, and *c* mean, and why are their accuracies different?)** The “API search” row in Table 3 corresponds to “Reference performance,” and has no connection to *n*, *l*, and *c*. Our intention was to restate that this reference performance was generated via API search. As noted in Table 3, *n*, *l*, and *c* are hyperparameters; we already direct readers to the Appendix and already incorporated a detailed explanation of mitigation methods and their hyperparameters in Appendix F.2 and F.3. Accordingly, these differences in hyperparameters lead to variations in accuracy. Nevertheless, to make this clearer, we have revised Table 3.
> - **Suggestion 5: Mathematical notation inconsistencies (D vs. \mathcal{D}) are confusing.)** We have fixed these inconsistencies and updated our *D* notation accordingly in the revised version. Specifically, we redefined the measure D_\vtheta, which predicts whether a prompt induces a memorized image, as d_\vtheta.
>
>
> [C1] Understanding and mitigating copying in diffusion models. NeurIPS 2024.
>
>
> [C2] On memorization in diffusion models. TMLR 2025.
>
>
> [C3] On the de-duplication of laion-2b. Arxiv preprint.
>
>
> **W2. Excluding dataset size, the advantage of the proposed benchmark over previous work [C4] is ambiguous. (+ Provide the prompt diversity & human evaluation)**
>
>
> As discussed in the related work section, our key difference from the previous work [C4] lies not only in the size of our dataset **but also in creating a standardized benchmark.** While the prior work [C4] aimed to search for diffusion-memorized images, its purpose is not on establishing a benchmark. This means that there were no defined metrics or methodologies for properly evaluating mitigation methods. In contrast, we provide a benchmark equipped with rigorous metrics, reference performance, and general prompt scenarios, enabling proper evaluation of mitigation methods. Reviewer `vYcx` also acknowledged our study for addressing the research gap, the lack of a standard benchmark.
>
>
> Nevertheless, in this rebuttal, we provide additional analysis to **highlight a key advantage of our dataset: the trigger prompts exhibit a wide range of difficulty levels (degree of memorization bias.)** In prior work [C4], trigger prompts were obtained by directly searching within the training data, resulting in highly biased prompts that almost always lead diffusion models to reproduce memorized images. This extreme bias can hinder accurate evaluation, as test sets composed solely of such challenging prompts may obscure the relative performance of different mitigation methods, especially when their strength is set as low. In contrast, our method searches trigger prompts from an open prompt space via MCMC, naturally leading to a diverse spectrum of memorization degree.  We have quantified and visualized this memorization bias distribution as a histogram in the revised version (Sec. 4.1, Fig. 1.(b)). We invite the reviewer to confirm these results.
>
>
> [C4] A Reproducible Extraction of Training Images from Diffusion Models, Arxiv preprint.

---

> > ### Author Response · Authors · 2025-04-12
> >
> > **W2-continued: for human evaluation,** we believe that the most critical aspect requiring human judgment is verifying whether the images generated from trigger prompts appear to be memorized. Both prior work [C4] and our benchmark follow the same structure of {trigger prompt, URL}, where the URL points to an image that is presumed to be memorized when the associated prompt is used for generation. In this context, the most important factor for researchers studying mitigation methods is whether the image at the provided URL is indeed memorized by the model. Therefore, we conducted human verification for this purpose. Specifically, for each trigger prompt, we randomly generated 20 images and evaluated whether any of them matched the image at the corresponding URL. As a result, we found that, among the 345 prompts provided by prior work, 40 failed to regenerate the target image—either because the image was not reproduced even after multiple generations, or because the URL was no longer accessible. In contrast, for all trigger prompts discovered by our method, the image at the corresponding URL was successfully regenerated, as we had already conducted human verification during the construction of the benchmark, as described in the paper. This suggests that the dataset from prior work may contain more noise, whereas our benchmark provides more reliable and reproducible trigger-image pairs. We believe this reliability makes our benchmark a more trustworthy resource for researchers developing memorization mitigation methods.
> >
> >
> > [C4] A Reproducible Extraction of Training Images from Diffusion Models, Arxiv preprint.
> >
> >
> > **W3. Analysis of the proposed benchmark is insufficient. Please provide  a model-wise analysis of the dataset statistics in Sec 4.1.**
> >
> >
> > Thank you for the valuable feedback. As mentioned in W1, we have incorporated it into Appendix B.3.
> >
> >
> > **W4. The deeper analysis of image memorization presented in Sec. 6 does not appear to be very fundamental. (Check theoretical works [C5])**
> >
> >
> > We appreciate the reviewer for presenting an excellent study. We have incorporated the suggested work into the related work section in the revised version of the paper. In contrast, please note that the purpose of Sec. 6 is not to offer a theoretical analysis of image memorization, but rather an empirical study of the memorization phenomenon. Reviewer `G77p` also mentioned that our findings in Sec. 6 are intriguing. We summarize the goals and findings of Section 6 as follows:
> > - We observed that the memorized images include actual commercially available products.
> > - Many empirical studies [C1, 2, 3] propose that image duplication is the primary cause of memorization. Yet, we found that even in LAION-5B—which has removed duplicates—image memorization still occurs. We conjecture that this may be attributed to layout duplication.
> >
> >
> > We hope this clarifies our intention to provide a deeper understanding of image memorization through these two observations.
> >
> >
> > [C5] Dynamical regimes of diffusion models, Nature Communications 2024
> >
> >
> > **W5. While measuring generation quality is important, I'm not fully convinced about replacing FID with Aesthetic Score—please provide FID for comparison.**
> >
> >
> > We have already provided each method's FID values evaluated on MemBench in Appendix E.1. These were originally omitted from the result section (Sec. 7) to keep the focus on method evaluation. In response to the reviewer’s request, we have added a note in Sec. 7 directing readers to Appendix E.1 for FID values.
> >
> >
> > **W6. Providing URLs might raise potential copyright and legal risks.**
> >
> >
> > Many image datasets such as LAION and Conceptual Captions are web-crawled and provide image URLs to users. In particular, the memorized images we searched are the subset of LAION, so we believe there is no significant issue. However, if there are further concerns raised by the ethical reviewer, we are willing to investigate this further.
> >
> >
> > We hope our answers address the reviewer's concerns. We would be happy to answer any further questions.

---

### Review · Reviewer_G77p · 2025-03-11

**Summary Of Contributions:**

This paper addresses the absence of a proper benchmark for evaluating how text-to-image generative models memorize training images. To remedy this, the authors introduce MemBench, a dataset containing several thousand “trigger prompts” that reliably produce memorized images. These prompts are generated and curated using Stable Diffusion, the masked language model BERT, and an MCMC sampling method—an approach that does not require direct access to the training set and is notably more efficient than naive baselines. As a result, MemBench gathers several times more trigger prompts compared to Webster (2023).

Furthermore, the paper leverages MemBench to examine existing memorization-mitigation methods, finding that while these methods attempt to reduce memorization, they also degrade text alignment and image quality. Consequently, the authors argue that further improvements are needed to address memorization in text-to-image generative models.

**Audience:**

Yes

**Broader Impact Concerns:**

The paper includes a Broader Impact Statement in the main text and provides detailed information about the dataset in the appendix through a data sheet section. As a result, there are no significant concerns regarding ethical implications that require further attention.

**Claims And Evidence:**

Yes

**Requested Changes:**

As mentioned in the weakness, verifying that trigger prompts can be generated and evaluated for models such as Pixart or Diffusion-DPO would further validate the method’s broader applicability.

**Strengths And Weaknesses:**

Strengths

- The authors efficiently generate and filter trigger prompts, outperforming naive baselines in terms of effectiveness and scalability.

- By evaluating existing memorization-mitigation techniques on MemBench, the paper provides valuable reference performance metrics for future research.

- The figures showing that Stable Diffusion memorizes commercial images and layout patterns are intriguing.

Weaknesses

- The current evaluation of trigger prompts focuses mainly on Stable Diffusion versions 1 and 2. While verifying memorization is impossible for some models, such as Stable Diffusion 3, there are still alternative options. For instance, applying the MCMC approach to models like Pixart [A]—trained on open-source SAM data [B] and using a T5 encoder—or Diffusion-DPO [C]—Stable Diffusion 1.5 fine-tuned on human preferences with pick-a-pic data—would illustrate broader applicability. Showing that trigger prompts can be extracted for such models and used to assess memorization-mitigation techniques would further validate the general utility of the proposed method.

[A] Chen et al., PixArt-α: Fast Training of Diffusion Transformer for Photorealistic Text-to-Image Synthesis, ICLR 2024.

[B] Kirillov et al., Segment Anything, ICCV 2023.

[C] Wallace et al., Diffusion Model Alignment Using Direct Preference Optimization, CVPR 2024.

---

> ### Author Response · Authors · 2025-04-12
>
> We appreciate the reviewer's feedback and thank the reviewer for recognizing both the technical contribution of our trigger prompt searching method and our work’s value as a benchmark.
>
>
> **W1. The trigger prompt searching method and evaluation were performed only on Stable Diffusion 1 and 2. Show that it also works on a broader range of models (e.g., Diffusion-DPO finetuned SD1.5).**
>
>
> First, please note that Table 2 in the main paper already presents the prompts identified for the [DeepFloydIF](https://huggingface.co/DeepFloyd/IF-I-M-v1.0) and [Realistic Vision](https://huggingface.co/SG161222/Realistic_Vision_V1.4), which shows the effectiveness of our proposed searching method.
>
>
> Meanwhile, **to show the scalability of our trigger prompt searching method** , we applied our algorithm to [Diffusion-DPO finetuned SD1.5](https://huggingface.co/mhdang/dpo-sd1.5-text2image-v1) and discovered trigger prompts. Within the limited time available for the rebuttal, we discovered over 20 new trigger prompts. For example, using the DDIM sampler, we confirmed that generating an image with the prompt *"dry bloodkrewred butcher rug"* resulted in the following three memorized images:
> - [Image 1](https://www.joom.com/en/products/647622f676278201ca8b8e3b)
> - [Image 2](https://luvlavie.com/collections/rugs/products/the-shining-rug)
> - [Image 3](https://www.joom.com/en/products/645379e54484500159a07028)
>
>
> Furthermore, **we evaluated the mitigation methods on the previously constructed DeepFloydIF dataset** using a limited set of hyperparameter configurations due to time constraints during the rebuttal period. The results were consistent with those observed in our experiments on Stable Diffusion 1 and 2. The results have been added to Appendix E.2.
>
>
> We hope this addresses the reviewer's concern regarding the general utility of our trigger prompt searching algorithm for evaluating mitigation methods. We would be happy to answer any further questions.

---

### Review · Reviewer_vYcx · 2025-03-30

**Summary Of Contributions:**

The paper introduces MemBench, the first benchmark for evaluating memorization mitigation methods in diffusion models. The authors highlight that diffusion models often regenerate memorized training images when triggered by specific prompts, raising concerns about copyright and privacy. MemBench addresses the lack of rigorous evaluation benchmarks by providing:

1. A large dataset of memorized image trigger prompts for multiple diffusion models (Stable Diffusion 1/2, DeepFloydIF, Realistic Vision).

2. Metrics (SSCD, CLIP Score, Aesthetic Score) to assess memorization mitigation methods.

3. A novel MCMC-based algorithm to efficiently search for trigger prompts without requiring access to training data.

4. Reference performance guidelines for mitigation methods.

Using the benchmark, the authors evaluate existing mitigation methods and find that current approaches significantly degrade model performance and image quality.

**Audience:**

Yes

**Broader Impact Concerns:**

No additional ethical concerns that would require expanding the Broader Impact Statement identified.

**Claims And Evidence:**

Yes

**Requested Changes:**

1. Illustration of how to ensure the trigger prompts found are semantically fluent. If not, please show some examples of the found prompts.

2. Add key findings of experiments in the introduction

3. More analysis of other types of the trigger prompts.

**Strengths And Weaknesses:**

Strengths:
1. The paper addresses a clear gap in the literature - the lack of a standard benchmark for evaluating memorization mitigation methods. The proposed benchmark and metrics are beneficial for the future development of image memorization scenarios.

2. The proposed MCMC method to quickly find the memorization trigger prompt is novel.

3. The proposed metrics are highly motivated.

Weaknesses:

1. The authors do not illustrate how to ensure the trigger prompts found are semantically fluent.

2.  It is recommended to highlight the key findings in the experiment in the Introduction, so the readers can better grasp the important conclusion from the paper.

3. More experimental analysis is expected. The authors need to carry out a deeper analysis to provide some valuable insights. For example, do authors explore more types of trigger prompts, e.g., a prompt of a landmark or a highly distinctive scene in a movie? I believe there are more types of trigger prompts can be incorporated.

---

> ### Author Response · Authors · 2025-04-12
>
> We appreciate the reviewer's valuable feedback. We especially appreciate the recognition of our benchmark’s contributions: (1) a large dataset, (2) rigorous metrics, and (3) reference performance, as well as our technical contribution in the MCMC searching method.
>
>
> **W1. Illustrate how to ensure the trigger prompts found are semantically fluent.**
>
>
> The semantic fluency is implicitly induced in our approach by leveraging BERT. In the last part of Sec. 3.3, we explain that the transition matrix for MCMC sampling is computed using the top-Q words predicted by BERT. The original purpose of this step is to enhance the computational efficiency of transition matrix calculation. However, it also induces semantic fluency. We have added an explanation of this point in Sec. 3.3 in the revised paper, and we thank the reviewer for highlighting this aspect.
>
>
> Additionally, although our method does not guarantee perfect semantic fluency, we believe that semantic fluency is not essential—and in some cases, prompts without semantic fluency may even be more valuable—for memorization mitigation. In the real world, those most concerned about memorization are likely the diffusion model providers. They would wish to avoid copyright-related risks and would not want their training data being extracted. However, a malicious user might attempt to reverse-engineer the training data using adversarial or abnormal prompts. To prevent such cases, it may be necessary to consider prompts that are not semantically fluent. Furthermore, since prompts without guaranteed fluency form a superset of semantically fluent ones, we believe they can contribute to the more robust mitigation methods.
>
>
> **W2. Highlight the key findings of the experiment in the Introduction.**
>
>
> We appreciate the suggestion. In the revised paper, we have expanded the “Benchmark Contribution” in the introduction section to elaborate on our key findings. We have also added a brief discussion about our empirical observations from Section 6: (1) some of the memorized images by diffusion models are of actual commercial products, and (2) memorization can arise from layout duplication.
>
>
> **W3. More analysis of types of trigger prompts.**
>
>
> We have additionally analyzed the types of trigger prompts. As previously noted in the paper, the memorized images generated by our trigger prompts can be categorized into three broad groups: commercial product, artwork, and human. Commercial products include phone cases, furniture, clothing, and baggage; artwork includes cartoons and brand logos; and human-related prompts include radio albums, actors, bands, movie posters, and soccer players. We have added these details to Section 4.1 in the revised paper.
>
> We hope our answers address the reviewer's concern. We would be happy to answer any further questions.

---

> > ### Comment · Reviewer_vYcx · 2025-04-30
> >
> > Thanks for the authors' response. The response has addressed my concerns.

---

### Author Response · Authors · 2025-04-12
**Response to all reviewers.**

We are grateful for the valuable feedback provided by the reviewers. Our paper introduces Membench, the first benchmark designed to evaluate memorization mitigation methods in diffusion models, along with an efficient algorithm to search for memorized image trigger prompts.


**Benchmark contribution acknowledged by reviewers:**
- MemBench provides a large number of trigger prompts (`vYcx`, `G77p`, `LXqR`) for various models (`vYcx`, `LXqR`).
- Comprehensive evaluation and rigorous metrics (`vYcx`, `G77p`, `LXqR`).
- Our first proposal of measuring an image quality score that previous works have overlooked (`vYcx`, `G77p`, `LXqR`).
- Reference performance that can offer guidelines for mitigation methods (`vYcx`, `G77p`).
- Empirical findings show that Stable Diffusion memorizes commercial images and that layout patterns induce memorization (`G77p`).




**Technical contribution acknowledged by reviewers:**
The reviewers acknowledged our innovative MCMC trigger prompt searching algorithm (`vYcx`, `G77p`, `LXqR`), particularly its scalability (`vYcx`, `G77p`), novelty (`vYcx`), and effectiveness even without training data (`G77p`, `LXqR`).




In this rebuttal, we provide the requested experiments and address all of the reviewers’ questions. We have incorporated the changes into the revised paper, which are marked in red for convenience.

---

### Decision · Action_Editor_B3Yp · 2025-05-02

**Recommendation:** Accept with minor revision

**Comment:**

The paper addresses a significant and acknowledged gap in the field by providing the standardized benchmark for evaluating memorization mitigation methods in diffusion models. The benchmark includes a substantial dataset of trigger prompts across various diffusion models, curated using a novel and efficient MCMC-based search algorithm that notably does not require access to the original training data. The proposed evaluation framework is comprehensive. The work provides valuable reference performance data for existing mitigation techniques, highlighting their current limitations. The benchmark and accompanying methodology are deemed meaningful, sound, and potentially highly useful for guiding future research.

Several presentation improvements were made. The paper now includes analysis quantifying the diversity of memorization bias in the discovered prompts and discusses human verification results comparing MemBench favorably to prior work in terms of reliability, strengthening the justification for the benchmark's quality beyond just size. Further analysis detailing the types of trigger prompts discovered and a model-wise breakdown of dataset statistics have been added.  All the above improvements  can be included in the final revised version.

**Audience:**

Diffusion models are a highly active and important area of research within the machine learning community. Understanding their properties, limitations, and failure modes, such as memorization, is crucial for researchers working on developing, improving, or applying these models.

**Claims And Evidence:**

The authors identify a critical gap in current research: the lack of standardized tools to assess how well mitigation techniques prevent models from regenerating training data images when triggered by specific prompts, a phenomenon with significant copyright and privacy implications. MemBench contributes a large dataset of such trigger prompts. Also, the work proposes a comprehensive evaluation methodology that assesses mitigation methods not only on their ability to reduce memorization for trigger prompts but also on their impact on the model's general performance and image quality. The paper also presents empirical findings suggesting memorization of commercial product images and the potential role of layout duplication as a cause for memorization. Te reviewers, after critically evaluating the paper, largely agreed that the claims were justified by the presented work, methodology, and experimental results. The evidence is convincing because it directly addresses the stated goals, uses sound methods (praised by reviewers), and leads to conclusions acknowledged as significant by the peer review process.

---

> ### Author Response · Authors · 2025-06-02
>
> Dear Action Editor, Reviewers, and Editors-in-Chief,
>
> We sincerely thank you for the insightful comments and careful evaluation that have greatly strengthened our paper. We have uploaded the camera-ready version, which now includes: **1)** a new analysis quantifying the diversity of memorization bias in the discovered prompts (Sec. 4.1), **2)** a model-wise breakdown of dataset statistics (Appendix B.3), **3)** additional explanations in the Introduction and Related Work sections, and **4)** writing refinements (clearer notation, improved captions, additional explanations, etc.).
> For completeness, we have also restored Datasheet details that were hidden during double-blind review. In addition, we now provide a public Github link to ensure transparency and reproducibility.
>
> We are grateful for the opportunity to contribute to the TMLR community and appreciate your support throughout the review process.
>
> Sincerely,
>
> The Authors.